# BUILDING SPATIAL WORLD MODELS FROM SPARSE TRANSITIONAL EPISODIC MEMORIES

**Zizhan He**[1,3]    **Maxime Daigle**[2,3]    **Pouya Bashivan**[1,2,3]
[1] Department of Computer Science, McGill University
[2] Department of Physiology, McGill University
[3] Mila, Université de Montréal

## ABSTRACT

Many animals possess a remarkable capacity to rapidly construct flexible cognitive maps of their environments. These maps are crucial for ethologically relevant behaviors such as navigation, exploration, and planning. Existing computational models typically require long sequential trajectories to build accurate maps, but neuroscience evidence suggests maps can also arise from integrating disjoint experiences governed by consistent spatial rules. We introduce the Episodic Spatial World Model (ESWM), a novel framework that constructs spatial maps from sparse, disjoint episodic memories. Across environments of varying complexity, ESWM predicts unobserved transitions from minimal experience, and the geometry of its latent space aligns with that of the environment. Because it operates on episodic memories that can be independently stored and updated, ESWM is inherently adaptive, enabling rapid adjustment to environmental changes. Furthermore, we demonstrate that ESWM readily enables near-optimal strategies for exploring novel environments and navigating between arbitrary points, all without the need for additional training. Our work demonstrates how neuroscience-inspired principles of episodic memory can advance the development of more flexible and generalizable world models.

## 1 INTRODUCTION

When visiting a new city, we quickly piece together scattered memories—an alley passed on a walk, a glimpse of a landmark from a taxi—into a mental map that lets us navigate, explore, and plan. This ability to build a coherent model of the world from fragmented experiences, and to use it flexibly for imagining new routes or outcomes, is a core feature of human intelligence that is fundamental to planning, problem-solving, and decision-making, all of which are crucial for survival (Bennett, 2023).

How do humans achieve this extraordinary feat? Insights from neuroscience and psychology provide compelling evidence for the neural mechanisms underpinning this capacity. One pivotal discovery is the dual role of a key brain area called the medial temporal lobe (MTL) in both spatial representations and episodic memories. Individual neurons within the MTL exhibit spatial selectivity, firing in response to specific locations (O'Keefe, 1976; Hafting et al., 2005), while collectively forming a population code that represents an animal's instantaneous position in space (Wilson & McNaughton, 1993). In parallel, MTL plays a critical role in the formation of episodic memories, enabling the rapid acquisition of associations between arbitrary stimuli (Howard & Eichenbaum, 2015). Strikingly, lesions to this area result in profound deficits: individuals lose not only their ability to explore and navigate effectively (Kimble, 1963) but also their capacity to form episodic memories (Scoville & Milner, 1957) and imagine novel scenarios (Hassabis et al., 2007). These converging findings strongly suggest that the MTL constructs structured relational networks, integrating overlapping episodic memories into a cohesive framework (Eichenbaum, 2004; McKenzie et al., 2014) (See Appendix B for more a detailed review). This mechanism may underlie a flexible and adaptive world model, allowing for inferences about spatial relationships beyond immediate perception.

Despite these insights, most prior computational approaches to world modeling have relied on learning from continuous sequences of recent observations and actions (Ha & Schmidhuber, 2018; Hafner

et al., 2019; Whittington et al., 2020; Levenstein et al., 2024; Bar et al., 2025; Fraccaro et al., 2018), fixed hand-designed circuitry for representing and updating spatial information (Wang et al., 2023; Chandra et al., 2025; Kymn et al., 2024), allocentric observations Khan et al. (2018), and iterative update procedures that hinders their ability to quickly adapt to environmental changes(Wang et al., 2023; Chandra et al., 2025; Stachenfeld et al., 2017). These models are typically trained on large datasets collected from a given environment, with knowledge encoded in their weights. While effective in certain contexts, such approaches face inherent limitations. For instance, agents may encounter different parts of an environment at widely separated times, visit states along arbitrary and unrelated trajectories that do not form continuous sequences, or experience environments that undergo structural changes (Fig. 1a). These challenges hinder the ability of sequence-based models to generalize effectively in environments where observations are disjoint and episodic.

This raises a key question: Can a neural network (NN) rapidly construct a coherent spatial map of the environment using only sparse and disjoint episodic memories? Drawing inspiration from the neuroscience of memory and navigation, we introduce the Episodic Spatial World Model (ESWM), a framework designed to infer spatial structure from independently acquired episodic experiences. ESWM meta-trains the model to predict unseen transitions from a sparse set of one-step transitions (referred to as the memory bank) sampled across diverse environments. This is in contrast with conventional world models in two ways: 1) ESWM does not require sequential observations for predictions–the memory bank can contain transitions from completely different episodes. This is crucial in large environments where a single walk spanning all locations might be long and therefore computationally expensive to process. In comparison, ESWM's memory bank can be constructed to contain only the useful transitions. 2) Each transition in the memory bank can be stored and updated independently. As a result, ESWM can rapidly adapt to environmental changes (e.g, addition of obstacles) by simply editing a handful of transitions that are related to that change (see Section 4.6).

Our main contributions are: 1) We introduce ESWM, a framework that models spatial environments from sparse, episodic transitions. 2) We find that ESWM's latent space forms a geometric map reflecting the environment's topology, which dynamically adapts to new memories and structural changes like obstacles. 3) We demonstrate that this learned model enables near-optimal, zero-shot exploration and navigation in novel environments without any task-specific training. 4) Finally, we show that ESWM is scalable to realistic continuous environments with high-dimensional inputs.

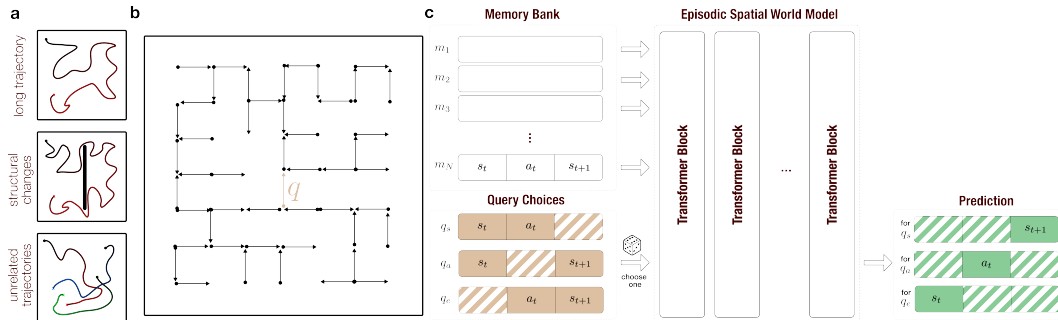

Figure 1: **Episodic Spatial World Model. a)** Three common scenarios that hinder the ability of typical world models to generalize effectively. (top) Observation of the full environment may take many time steps leading to long sequences; (middle) Environment structure may change across trials; (bottom) Information about a particular environment may be collected across separate exposures to the environment and not within a single one. **b)** Memory bank and query selection in a square grid environment. **c)** Architecture of ESWM and training procedure. Model input consists of a bank of transitional memories (corresponding to the black arrows in (b)) and a single query (q arrow in (b)), with either start-state, action, or end-state randomly masked with equal chances. The sequence of transitions is processed by a sequence model (e.g, Transformer encoder block) and the model parameters are updated to output the correct value for the masked component.

## 2 RELATED WORKS

### 2.1 WORLD MODELS

Humans excel at forming mental models of the world from egocentric observations and action, enabling efficient interaction with complex, dynamic environments. Inspired by this cognitive ability, prior works have used NN to forecast future outcomes based on past and present states (Ha & Schmidhuber, 2018) using methods such as image reconstruction (Watter et al., 2015; Alonso et al., 2024) in domains like video games (Hafner et al., 2020; Schrittwieser et al., 2020; Kaiser et al., 2019) and robotic control tasks (Gupta et al., 2022; Ebert et al., 2018). In spatial settings, agents have been trained to predict future sensory observations following arbitrary actions, typically using recurrently-connected networks (Stachenfeld et al., 2017; Whittington et al., 2020; Levenstein et al., 2024; Fraccaro et al., 2018).

More recently, the emergence of transformer architectures (Vaswani et al., 2017) has opened new possibilities for improving world models. Recent works such as TWM (Robine et al., 2023), STORM (Zhang et al., 2024), and IRIS (Micheli et al., 2022) have leveraged transformers-based world models to improve performance on sample-limited benchmarks like Atari 100k. However, unlike our proposed model, these models train extensively on a single environment and absorb the knowledge about that environment in the model's weight.

Most notably, our work is closely related to Generative Temporal Models with Spatial Memory (GTM-SM) Fraccaro et al. (2018) and the Tolman-Eichenbaum Machine (TEM) (Whittington et al., 2020; 2022), the latter of which has been proposed as a model of medial temporal lobe function. These models factorize sensory and structural information. The shared structure is encoded in a recurrent network weights, while environment-specific observations are stored in Hebbian-slot-memory, self-attention module, or differentiable memory dictionary. While both of these model classes include content addressable memory modules, important distinctions separate them from our proposed model. First, GTM-SM and TEM both presuppose a common structural template across environments (e.g, an open 2D grid), whereas ESWM does not. Second, unlike these models, ESWM does not embed structural knowledge in its weights but instead infers it dynamically from external memories, enabling it to capture distributions of environments with varying structures (e.g, a set of 2D mazes) where these models fail (Fig. 2). Finally, while GTM-SM and TEM operate over sequential trajectories, ESWM integrates disjoint experiences, making it more sample-efficient and better suited to rapid adaptation, such as handling newly introduced obstacles. See Appendix A for a more comprehensive comparison with other MTL models.

### 2.2 NAVIGATION AGENTS

The concept of cognitive map, an internal allocentric representation of the space, O'keefe & Nadel (1978) has been thought to enable navigation and spatial reasoning. Several studies have demonstrated that similar spatial representations emerge in NN trained on navigation tasks (Gornet & Thomson, 2024). For example, Banino et al. (2018) and Cueva & Wei (2018) showed that spatial representations arise in recurrent neural networks trained for path integration, while in parallel, NN lacking explicit mapping modules have achieved high performance on navigation tasks in unseen environments (Wijmans et al., 2019; Reed et al., 2022; Khandelwal et al., 2022; Shah et al., 2023). Wijmans et al. (2023) recently showed that spatial representations can emerge in navigation-agnostic NN architectures trained for navigation, emphasizing memory's role in shaping these spatial representations. This raises a key question: *Can spatial maps also emerge in general-purpose models not explicitly trained for navigation and operating entirely on a finite set of episodic memories?*

## 3 METHODS

### 3.1 EPISODIC SPATIAL WORLD MODEL

Formally, our Episodic Spatial World Model (ESWM) is a function $f$ that takes as input a memory bank $M$ and a partially masked transition $q$, and predicts the complete transition $q^*$:

$$q^* = f(M, q) \tag{1}$$

We define an episodic memory as a one-step transition in a spatial environment, represented by the tuple $(s_s, a, s_e)$—denoting the source state, action, and end state, respectively. For each environment, we construct a memory bank $M$ as an unordered set of such transitions, constructed to be: *Disjoint*, transitions do not form a continuous trajectory; *Spanning*, the transitions collectively form a connected graph covering all locations; *Minimal*, removing any transition would disconnect the graph (Fig. 1b). However, *Minimality* is not required for training ESWM but is imposed in the grid environment to prove that the absolute minimum information is sufficient to construct a functional spatial map (see Fig. 9 for results on non-minimal memory banks). We algorithmically generate such memory banks to sparsely cover each environment (Extended Methods C.3). Importantly, a single room may yield multiple valid memory banks, enabling our meta-training setup.

We formulate spatial modeling as the ability to infer any missing component of an unseen but plausible transition $q$—that is, a query transition not present in $M$ (Fig. 1b). The task is to predict the masked element (either $s_s$, $a$, or $s_e$) of $q$, given the other two and the memory bank $M$ (Fig. 1c). For example: predicting the next state from $s_s = 5$ and $a =$ move right; inferring the action connecting states 5 and 10; or recovering the start state given $a =$ move right and $s_e = 10$.

To train ESWM, we adopt a meta-learning approach where on each trial (i.e. sample), we randomly sample: (1) An environment $e$ (though random assignment of states to locations); (2) A disjoint memory bank $M$ from $e$; (3) A plausible query transition $q$ from $(e \setminus M)$; (4) A masking choice (randomly masking $s_s$, $a$, or $s_e$). This process prevents the model from memorizing any specific environment since the state-location mapping is randomized across trials (i.e, the model cannot simply memorize that state "5" is at top-left because "5" could be at top-right the next trial), and our environments (Open Arena, Random Wall; detailed below) contain an intractable amount of configurations ($> 10^{33}$, see Extended Methods C.2, Fig. 2).

## 3.2 Model Selection and Training

We considered three NN architectures for $f$, 1) Encoder-only transformer with no positional encoding(Vaswani et al., 2017), ESWM-T, 2) long short-term memory (LSTM) (Hochreiter & Schmidhuber, 1997), ESWM-LSTM, and Mamba (Gu & Dao, 2024), ESWM-MAMBA. Each transition component $(s_s, a, s_e)$ is projected separately into a shared high-dimensional space and averaged to form a single input token.

The transformer model receives as input the concatenated sequence of memory bank and query tokens (Fig. 1b). The LSTM and Mamba models receive a randomly permuted sequence of memory tokens, each with the query token added. Three separate linear heads read out the model's prediction on the three parts $(s_s, a, s_e)_q$ from the final layer's last token. We use cross-entropy loss between predicted and actual states and actions. Equal weight is assigned to each prediction $(s_s, a, s_e)$. Training is done over 460k iterations with a batch size of 128 and a cosine learning rate schedule.

A spatial world model that assumes complete memory coverage is rather constrained. Instead, it should reason over partial observations. To achieve this, we create trials where some memories are deliberately removed. The model is expected to solve queries within observed regions and classify unsolvable ones (e.g., transitions involving unobserved regions) under an additional "I don't know" category (Fig. 2b).

## 3.3 Model Evaluation

We evaluate ESWM under three settings of varying complexity:

1. *2D grid environments:* We evaluate ESWM in a family of discrete, hexagonally structured grid environments (Extended MethodsC.2). Each location in the grid environment is assigned an integer-valued observation, drawn from a predefined set of states. An environment is uniquely defined by its mapping from states to spatial locations, allowing us to generate a vast number of distinct rooms by randomly shuffling this mapping—critical for our meta-learning objective. The hexagonal layout allows for six possible actions per location, offering richer connectivity than traditional square grids. We design two types of environments to test generalization: (1) Open Arena to evaluate generalization to unseen states and; (2) Random Wall to evaluate generalization to unseen structures.

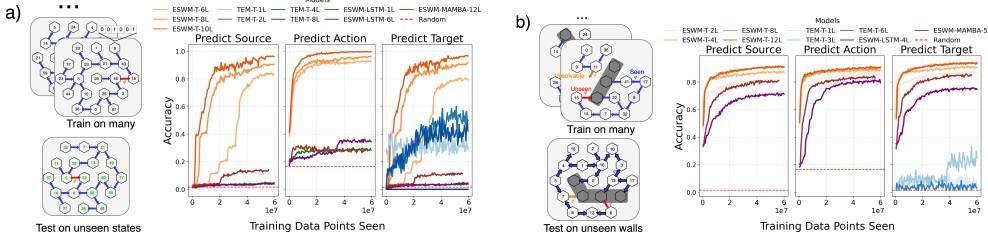

Figure 2: **Evaluation accuracy in a) Open Arena and b) Random Wall**. In both schematics, blue arrows are transitions in the memory bank and the $q$-labeled arrows are examples of query transitions $q = (s_s, a, s_e)$. In `Open Arena`, the states are represented as 6-bit binaries and the model is tested in environments entirely filled with unseen states, while in `Random Wall`, the states are integers and the model is tested on its generalizability to unseen wall patterns. In addition, a random subregion is masked in `Random Wall` and the query $q$ can be either unsolvable, unseen, or seen, depicted as $q$-labeled arrows with different colors. The `Random Wall` displayed here is a scaled down version of what the model is trained on (19 v.s 36 locations).

- `Open Arena`: Each room consists of 19 locations and 64 possible state values. These states are represented compositionally as 6 binary bits (Fig. 2a), enabling us to train and test ESWM on *mutually exclusive sets of states* (e.g, train on odd states and test on even states). For action prediction, the model performs 6-way classification (one for each action). For source and end state prediction, the task is broken into six independent binary classification problems—one for each bit. This setup forces it to build representations that support spatial reasoning even in environments filled with previously unseen but compositionally generated state observations.
- `Random Wall`: To test for generalization to unseen structures, this environment features larger rooms (36 locations) with unique integer states and a randomly shaped wall of obstacles (Fig. 2b). During testing, the model must generalize to novel wall configurations and state-location mappings. The prediction task is 6-way classification for actions and 36-way for states. We simplified the observations to integers to focus the task on inferring the environment's underlying structures.

2. *MiniGrid environments:* We used MiniGrid (Chevalier-Boisvert et al., 2023) to create environment layouts of size $9 \times 9$ with egocentric multi-dimensional observations and actions (Fig. 6). Each state consist of an egocentric $5 \times 5$ view of the environment containing non-unique colored objects. The agent takes one of three egocentric actions: turn left, turn right, go straight. Each procedurally generated environment is assigned a random color layout and wall pattern. The model has to self-localize based on partial egocentric views of the environment experienced from varying head directions.

3. *Simulated 3D indoor environments:* We use ProcThor (Deitke et al., 2022), a large set of procedurally generated 3D indoor scenes. Each state consists of a high-resolution RGB first-person observation ($224 \times 224$). The agent takes action defined by egocentric displacements $\Delta xy$ (along left/right and front/back axes) and rotation $\Delta \theta$, both from a continuous range.

For MiniGrid and ProcThor environments, the memory bank consists of episodic memories that collectively cover the room, with partial overlap across observations to enable memory integration. Importantly, the memory banks in these environments remain compact.

## 4 RESULTS

### 4.1 MODELING SPATIAL ENVIRONMENTS FROM EPISODIC MEMORIES

We first assessed ESWM on its core ability to predict missing elements of unseen transitions given a set of memories from an environment and a partially observable query. In both settings, we consider a closely related sequence-based model TEM-T as a baseline (Whittington et al., 2022). Instead of a set of disjoint memories, TEM-T receives a spanning trajectory as the memory bank and is queried

to predict an unseen transition at the end of the trajectory. To enable fair comparison, the spanning trajectory is generated to be minimal (Extended Methods C.5).

We found a stark contrast between the model types in `Open Arena` setting (Fig. 2a). The ESWM-T model excelled at learning this task while ESWM-LSTM and ESWM-MAMBA models overfitted (see Fig. 8) and struggled to reach above chance on the validation set. This suggests that the attention mechanism in Transformer architectures—reminiscent of classic content-addressable memory models (Kanerva, 1988)—plays a key role in learning generalizable world models from episodic memories. However, simpler, hand-engineered memory-lookup mechanisms like those in Coda-Forno et al. (2022a) fail on this inference task (Fig. 17), highlighting the need for learnable attention-based mechanisms. Moreover, we also show that ESWM-T can also model spatial environments when trained on inputs distributions that lack the *Minimality* constraint (Fig. 9; see section D.2 for comparing ESWM-T trained with/without the *Minimality constraint*). In addition, the sequence-based transformer model TEM-T, while outperforming non-transformer ESWM models, still significantly underperforms relative to the transformer ESWM-T, highlighting the advantage of directly inferring from episodic memories, rather than segregated encoding of structure and memories. Furthermore, while ESWM-T clearly improves with larger model size, the performance in larger TEM-T models significantly degrades (Fig. 2a-b). For further discussions in this setting, see section D.1.

In the second scenario, the `Random Wall`, we found that all ESWM models are capable of learning the task (Fig. 2b). Although, there was a substantial gap between transformer-based models and other models, even with equal number of parameters (LSTM-4L, T-2L, and MAMBA-5L). This showed that ESWM learned a general strategy for modeling an environment from sparse and disjoint episodic memories, enabling it to generalize across environments with different structures. Consistent with our prediction, TEM-T fails at modeling the set of environments with varying structures. This underscores the importance of direct structural inference from memories rather than encoding it in the model's weights.

Finally, we also considered environments with duplicate (non-unique) states. In the absence of historical context, one-step transition prediction prevents the model from disambiguating between duplicate states. However, this limitation can easily be overcome by prepending a trajectory, serving as context, to the query. To show this, we train ESWM-T in environments with duplicated states and find that it is capable of solving the task in both `Open Arena` and `Random Wall` (see Fig. 7).

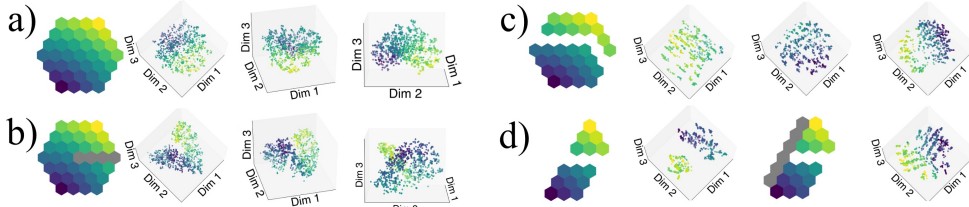

Figure 3: **Spatial map emerges in ESWM's latent space. a), b)** ISOMAP projections of ESWM-T-2L's activations for action prediction. Different columns are the same projection from different viewing angles. **a)** Spatial map in the absence of obstacles and **b)** in the presence of obstacles (a straight wall). **c), d)** ISOMAP projections of ESWM-T-14L's activations. **c)** From left to right: A sample room with two disconnected regions whose shapes match boundaries' shape; ESWM's latent space when a memory bank observing either the top or bottom region is given as input; ESWM's latent space when a memory bank observing both regions is given as input. **d)** From left to right: A sample room containing two disconnected regions whose shapes give no cues about their relative position; ESWM's latent space when a memory bank of the room is given as input; Updated room with a new wall, the regions remain disconnected; ESWM's latent space when a memory bank of the updated room is given as input. See Fig. 10– 14 for more examples.

## 4.2 Spatial Maps Emerge Robustly Within ESWM's Latent Space

To simulate an unseen transition, ESWM must integrate disjoint memory fragments into a coherent internal map that reflects both observed states and obstacles. Inspired by low-dimensional, environment-like neural manifolds in rodents' MTL Nakai et al. (2024), we applied ISOMAP Tenen-

baum et al. (2000) (Extended Methods C.6) to ESWM's latent space. The resulting manifolds revealed that ESWM consistently organizes its internal representations to mirror the spatial layout of each specific environment.

In obstacle-free environments, the manifold adopts a smooth, saddle-like shape (Fig. 3a). When obstacles are introduced, they create localized discontinuities that correspond precisely to blocked regions in physical space (Fig. 3b). This structure is remarkably robust and persists under out-of-distribution obstacle shapes (Fig. 10, 11), partial observation of environments (Fig. 12), across architectures (LSTM, Transformer, Mamba; Figs.10–11), and throughout all transformer layers, which become progressively more continuous with depth (Fig. 14). All prediction tasks ($s_s$, $a$, $s_e$) yield similar maps (Fig. 12a). In the presence of disconnected memory clusters with unclear spatial relations, ESWM pieces its fragmented internal representation into a coherent map by aligning them along the inferred wall shapes or boundaries' shapes (Fig. 3c,d; Additional Analysis D.3). Quantitatively, path lengths in latent and physical spaces correlate strongly ($R^2 = 0.89$, $N_{\text{path}} = 1500$, $N_{\text{env}} = 75$; Fig. 16b), and a linear classifier on ESWM activations can accurately discriminate which of two states is farther from a reference state with $93.42 \pm 0.006\%$ accuracy (chance = 50%, $N_{\text{seed}} = 10$; Extended Methods C.7). Furthermore, the sRSA (Levenstein et al., 2024) of a trained ESWM is significantly higher than that of an untrained one (trained $= 0.78 \pm 0.02$, untrained $= 0.5 \pm 0.04$, $n_{\text{env}} = 1000$; paired $t$-test, $t$=172.17, $p < 10^{-3}$; Extended Methods C.8).

Notably, such structured spatial representations do not emerge in episodic agents trained for navigation, such as Episodic Planning Network (EPN; Ritter et al. (2020)), despite being trained with meta-RL on the same environments and data as ESWM (Fig. 10, 11; Appendix H).

### 4.3 ESWM Integrates Overlapping Memories

We ran several experiments to test whether ESWM truly retrieves the information scattered across its memory bank. First, we find that ESWM's prediction uncertainty—measured by entropy of its output distribution probability—increases with the length of the integration path needed to solve a query (Fig. 4a). Second, we found that introducing shortcut memories that reduce this path length, significantly alters the prediction distribution (independent two-sample t-test; $n = 5000$, $p < 0.001$; Fig. 4b). Third, we find that ESWM trained in size-19 environments can generalize to substantially larger environments of size 37 in `Open Arena`, achieving lower, but still well-above-chance prediction accuracy ($s_s$: 59%, $a$: 90%, $s_e$: 55%)– unlikely if the model had simply memorized patterns. Fourth, ESWM's performance improves with denser, non-minimal memory banks, despite being out-of-distribution in `Random Wall` (Fig. 4c). Lastly, despite being trained only on clean data, ESWM develops some robustness to noisy memory banks. Its prediction accuracy only dropped by 1% and 10%, respectively, when tested on memory banks containing contradictory state observations and transition outcomes ($n$=5000; Extended Methods C.9). Altogether, these results suggest that ESWM indeed builds coherent spatial representations by integrating overlapping episodes.

### 4.4 Exploration in Novel Environments

So far, we have assumed that ESWM has access to informative episodic memories. However, animals actively acquire such memories through exploration. The ability to rapidly map out novel environments—identifying obstacles, rewards, and threats without prior knowledge—is critical for survival. We show that ESWM enables autonomous exploration in unfamiliar spaces without any additional training.

We designed a simple exploration algorithm based on ESWM's "I don't know" predictions (Alg.2). Starting with an empty memory bank, the agent evaluates all possible actions at each step by querying ESWM. It selects the action with the highest predicted uncertainty (i.e., highest "I don't know" probability) and adds the resulting transition to memory. When ESWM is confident about all immediate actions, the agent performs multi-step lookahead by unrolling the model to identify frontier states with low prediction confidence (Yamauchi, 1997). A prediction is deemed unconfident if ESWM either outputs "I don't know" or assigns $\leq 80\%$ probability to the predicted outcome.

Interestingly, the agent adopts an efficient zig-zag exploration strategy, only revisiting states when separated by long paths. In the `Random Wall` environment, ESWM explores 16.8% more unique

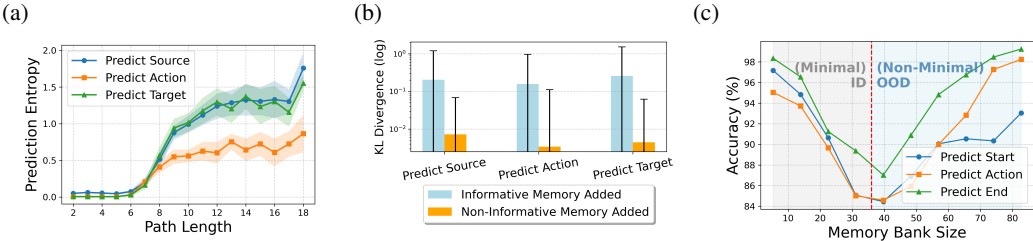

Figure 4: **ESWM integrates memories.** **a)** X-axis is the shortest path length between the source and end states in the query, with edges corresponding to episodic memories in the memory bank, and the Y-axis is the entropy of ESWM's prediction probabilities (n=5000).The shaded region is 95% confidence interval computed over all predictions for each path length. **b)** KL divergence between ESWM prediction distributions before and after adding an episodic memory to the memory bank. An informative episodic memory shortens the integration path required for the model to solve the prediction task, while a non-informative episodic memory does not (n=5000). **c)** Prediction accuracy across memory bank sizes. In the gray region, memory banks minimally observe the environment; larger banks correspond to larger observed areas. In the blue region, the observed area is fixed (full environment), so larger memory banks correspond to a more densely observed area. ID and OOD denote in-distribution and out-of-distribution settings. (n=2000). ESWM-T-14L is used for a) and b) while ESWM-T-12L is used for c). All experiments are in `Random Wall`.

states than EPN (one-way ANOVA; $n = 1000$, $F = 2803.63$, $p < 0.05$; Tukey's HSD post hoc test: ESWM > EPN, $p < 10^{-3}$), and achieves 96.48% of the performance of an oracle agent with full obstacle knowledge (Fig. 5a, Fig. 15). When exploration continues until the memory bank reaches the max length seen during training, ESWM visits on average $91 \pm 2\%$ of all unique states ($n_{trials} = 1000$). Moreover, the collected memory banks allow ESWM to predict $s_s$, $a$, $s_e$ with 97%, 93%, 98% of the accuracy of those achieved using procedurally generated memory banks ($n = 1000$). Together, these results highlight ESWM's ability to support data-efficient, near-oracle exploration and mapping in novel environments.

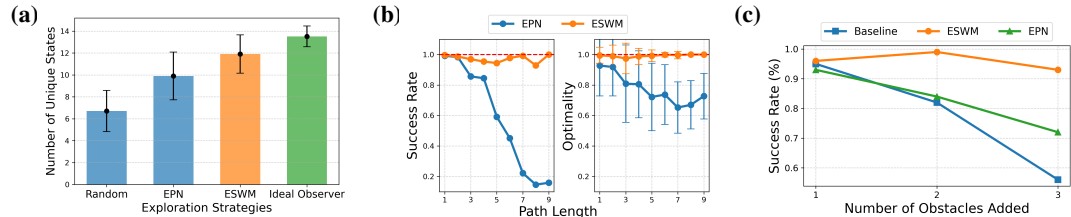

Figure 5: **Exploration, navigation, and adaptability.** **a)** Comparison of exploration strategies based on the number of unique states visited over 15 time steps in Random Wall. The optimal agent explores along the path found by the Traveling Salesman Algorithm over known free space. N=1000. **b)** Comparison of navigation success rate and path optimality over path lengths between EPN and ESWM, n=2400. **c)** Comparison of navigation success rates with increasing number of unexpected obstacles. The baseline agent is trained on the original environment and tested after structural changes, while ESWM navigates with memory banks from the original environment and needs to autonomously adapt to changes (n=100). The best seed for ESWM and EPN is used. We compare their performance across seeds in
Fig. 15. Random Wall Experiments include 19 locations.

## 4.5 NAVIGATING WITH MINIMAL SPANNING EPISODIC MEMORIES

A key benefit of a world model is the ability to plan by simulating trajectories. We test this capability by using the memory-bank-equipped ESWM for planning in the `Random Wall` environment. The task is to find a path from a source state $s_{\text{source}}$ to a goal state $s_{\text{goal}}$ using only the current observation and the episodic memories in the memory bank, without access to global information like coordinates or a complete map of obstacles.

Our planning strategy relies on ESWM's ability to infer the end state of any given transition. Initialized at a state, an agent can imagine multiple steps into the future by querying ESWM on the consequences of all applicable actions, bootstrapping on ESWM's own predicted end states, and tracking visited states until the goal state is reached (Alg.1). Then the agent can act out the imagined sequence of actions in reality.

Using this strategy, an ESWM-based navigational agent can navigate between arbitrary states in `Random Wall` with 96.8% success rate and 99.2% path optimality, outperforming EPN Ritter et al. (2020) by 18% and 21% respectively (Fig. 5b) (one-sided independent t-test; n=2400: ESWM>EPN on optimality; $p < 10^{-3}$). Navigation is successful if the agent reaches the target within 15 time steps, and path optimality is defined as the ratio between the shortest and actual path length. We also include a variant of such an algorithm in the appendix (Additional Analysis D.4) that biasedly considers the action that shortens the geodesic distance to the target on ESWM's latent spatial map, demonstrating how structured latent spaces in world models can serve as efficient heuristics for planning.

It is important to emphasize that the navigation success primarily stems from the high-fidelity world models constructed by ESWM using minimal memory (requiring $4\times$ fewer memories than EPN), rather than from the planning algorithm itself. Thus, the key finding is not that a classic search algorithm can outperform an RL agent, but that a world model learned from such sparse, disjoint data is accurate enough to enable it to do so.

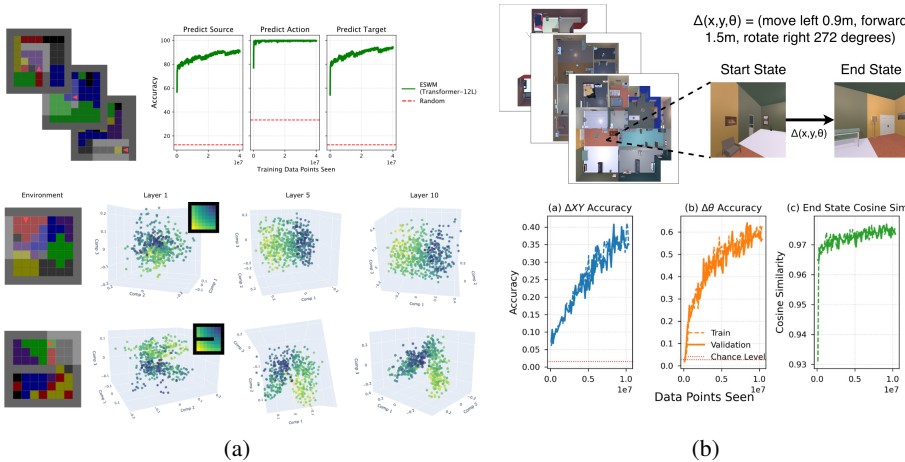

Figure 6: **ESWM in scaled-up environments a)** (Top-Left) Examples of procedurally-generated MiniGrid environments. (Top-Right) Accuracies for $s_s$, $a$, and $s_e$ predictions for ESWM-T-12L. (Bottom) ISOMAP analysis on ESWM's latent space shows a map-like representation. Inset shows the color scheme across spatial locations. **b)** (Top-Left) Examples of procedurally-generated Proc-Thor environments. (Top-Right) A sample transition. States are high-dimensional images, and actions are continuous 3-dimensional vectors. (Bottom) ESWM-T-12L's performance in unseen test environments for action (measured by accuracy on sub-components $\Delta xy$ and $\Delta \theta$) and end-state prediction (measured by cosine similarity to the frozen vision model's representation).

## 4.6 ESWM ENABLES FAST ADAPTATION TO CHANGES

Unlike traditional world models that embed environment knowledge in their weights, ESWM decouples memory from reasoning. This separation makes ESWM inherently adaptable to changes: when the environment changes, no retraining is needed—only the external memory bank is updated. Because ESWM operates on sparse, independent one-step transitions, targeted edits to memory are straightforward. We demonstrate this flexibility in a dynamic navigation task. After introducing substantial changes—such as new obstacles—an ESWM agent continues to navigate between arbitrary states with a 93% success rate (Fig. 5c).

The adaptive agent works as follows: Given a (possibly outdated) memory bank $M$, it plans a path from $s_{\text{start}}$ to $s_{\text{goal}}$ (Section 4.5). During execution, it compares predicted observations with actual ones. When mismatches occur (indicating environmental changes), the agent removes outdated transitions (where $s_s$ or $s_e$ match the predicted observation from $M$), prompting exploration (Section 4.4) to fill the gaps. After acquiring new transitions, the agent replans with the updated memory.

ESWM's adaptability significantly surpasses both EPN and a baseline RL agent trained specifically for the original environment (see Extended Methods C.10 for baseline details). For example, after adding a new wall consisting of multiple contiguous obstacles, ESWM maintains a 93% navigation success rate, whereas EPN and the baseline drop to 72% and 56% respectively (see Fig. 5c). Unsurprisingly, the baseline agent fails to form any structured latent spatial map.

### 4.7 SCALING ESWM

In the previous sections, we focused on simple grid environments, where episodic memories are transitions between integer observations via allocentric actions (e.g., go north), to examine how ESWM behaves under various scenarios. To test ESWM's scalability, we next evaluate it in Minigrid and ProcThor (see Section3.3)–two more complex scenarios with egocentric high-dimensional observations and actions. To adapt ESWM to MiniGrid and ProcThor, its operation on a discrete bank of transitions remains unchanged, and we only adjusted the encoding function for states (trained from scratch for MiniGrid, pretrained vision model for ProcThor) and actions and the readout function for generating the predictions to match those in this environment (Extended Methods C.1).

In MiniGrid, ESWM continues to exhibit strong predictive performance (Fig. 6a) Top-Right) while constructing coherent internal map representations (Fig. 6a) Bottom) that adapt to the environment's structure. In ProcThor, our results show that ESWM is solving this challenging task by learning to predict both state vectors and action parameters (Fig. 6b).

## 5 DISCUSSION

In this work, we introduced ESWM, a NN model that can rapidly construct a coherent spatial world model from sparse and fragmented one-step transitions. ESWM exhibits strong spatial reasoning and supports downstream tasks like exploration and navigation—even without explicit training for them. Unlike prior models that rely on fixed circuitry for spatial representation (Wang et al., 2023; Chandra et al., 2025; Kymn et al., 2024), ESWM learns both representation and update mechanisms directly from experience. It encodes new memories in a single shot, avoiding the iterative parameter tuning required by earlier approaches (Wang et al., 2023; Chandra et al., 2025; Stachenfeld et al., 2017). By storing transitional memories instead of state-observation pairs (Whittington et al., 2020; Coda-Forno et al., 2022b; Whittington et al., 2022), ESWM also adapts quickly to environmental changes, such as the addition or removal of obstacles—an area where many existing models struggle.

A limitation is the use of highly controllable environments, which allowed us to scale up the training and control different aspects of the environment and data freely. Future work could explore extending this approach to more realistic settings such as robotic navigation with natural, complex sensory information. Although theoretical, our work has potential societal impacts in autonomous navigation and exploration, highlighting the need for safe, ethical, and responsible deployment (see Section I).

## 6 REPRODUCIBILITY STATEMENT

We provide detailed descriptions to ensure the reproducibility of our work. These include training details (Extended Methods C.1, data generation pipeline (Extended Methods C.2 - C.4), baseline implementations (Extended Methods C.5, C.10, Section H), methods of analysis (Extended Methods C.6, C.7), and algorithm pseudo codes (Section G).

## 7    DECLARATION OF THE USE OF LLM

We recognize the importance of LLM usage disclosure and declare that our use of LLM is strictly limited to word editing and polishing.

## 8    ACKNOWLEDGEMENT

M.D. was supported by the UNIQUE PhD Fellowship. This research was supported by the NSERC Discovery grant RGPIN-2021-03035, and CIHR Project Grant PJT-191957. P.B. was supported by FRQ-S Research Scholars Junior 1 grant 310924, FRQNT-NSERC NOVA grant 2024-NOVA-346823, and the William Dawson Scholar award. All analyses were executed using resources provided by the Digital Research Alliance of Canada (Compute Canada) and funding from Canada Foundation for Innovation project number 42730.

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

APPENDIX

## A   RELATION TO OTHER MODELS OF MEDIAL TEMPORAL LOBE

**Successor Representations (SR).** In the context of reinforcement learning, successor representation has been shown to explain many non-spatial aspects of hippocampus responses (Stachenfeld et al., 2017). SR and ESWM are similar in that both rely on transition information, but they differ in several important ways. First, classical SR methods require either a full transition matrix or incremental experience sufficient to estimate one. They are not designed to infer missing transitions or to complete a partially observed graph. ESWM, by contrast, is built specifically for that setting: it takes a sparse set of one-step episodic transitions and infers the missing structural relations directly at inference time. Conceptually, ESWM is estimating the entire family of transition distributions from incomplete data, whereas SR presupposes access to the full underlying dynamics.

Second, SRs are learned through iterative, policy-dependent updates and encode multi-step, discounted future occupancy. ESWM does not rely on iterative updates and is not tied to a particular policy. The transitions in its memory bank are single, policy-free episodic observations, and the model performs transitive reasoning over these episodes on demand to answer arbitrary transition queries.

In addition, SRs collapse the environment into a future-occupancy representation, which mixes structural and predictive information. ESWM keeps the structure explicit and manipulable, allowing it to reconstruct the transition graph itself rather than a discounted predictive summary (e.g. state values). ESWM is also symmetric in its inference capabilities: it jointly estimates forward transitions, backward transitions, and actions from the same memory bank, whereas SRs typically encode only forward predictive relationships.

Finally, ESWM supports rapid structural updates by directly editing or adding episodic transitions, without needing to relearn a transition matrix. This contrasts with SRs and related models, which must re-estimate transition dynamics when the environment changes.

**Tolman-Eichenbaum Machine.** Tolman–Eichenbaum Machine (TEM)(Whittington et al., 2020; 2022; Bakermans et al., 2025) has been proposed as a model of medial temporal lobe function and is meta-trained across diverse environments to predict unseen transitions from episodic experience. TEM factorizes sensory and structural information: the shared structural scaffold is encoded in the weights of a recurrent network, while environment-specific observations are bound to that scaffold through a Hebbian-slot memory or self-attention module. The recurrent state therefore encodes a generic structural prior—typically a Euclidean, grid-like topology—and the attractor dynamics allow for content-specific retrieval. This makes TEM well-suited for environments that share this underlying family of structures and allows it to tolerate large changes in sensory input at familiar locations. However, this same design makes the model less flexible when the underlying topology itself changes or when the environment obeys different structural rules (e.g., RandomWall environments where walls can appear in arbitrary shapes and configurations).

In contrast, ESWM does not factorize structural and sensory information. Both the structural layout and the expected sensory observations are inferred jointly and directly from episodic transitions. Because ESWM does not rely on a fixed structural template encoded in its weights, it can accommodate changes to the environment by directly modifying its memory bank–allowing it to update predictions immediately without retraining. This makes ESWM particularly effective in settings where the topology can vary widely across environments or can change abruptly within an environment.

Finally, whereas TEM operates on sequential trajectories and depends on an explicit dynamical update of its recurrent state, ESWM integrates disjoint and non-sequential experiences. This enables more sample-efficient learning from sparse observations and supports rapid adaptation to newly introduced obstacles or structural changes that were never part of a trajectory.

**Recurrent Predictive Models**. Recurrent predictive models have also been proposed as models of neural computations within the hippocampus (Levenstein et al., 2024). These models consist of recurrent neural networks trained to predict future observations from a sequence of prior observations along a particular trajectory. In these models, both the structural and sensory information are em-

bedded directly in the recurrent network's weights. This architecture makes them even less flexible in the face of changes, more so than the TEM class of models. Whereas TEM includes an episodic module that allows for rapid adaptation to sensory changes within a fixed structural template, recurrent predictive models must update their recurrent weights to incorporate both sensory and structural changes, which is considerably harder to achieve quickly.

In addition, recurrent predictive models depend on having continuous, sequential experience to gradually shape the internal dynamics into a coherent predictive structure. This reliance on dense experience makes them poorly suited to integrating disjoint or fragmentary observations. ESWM, by contrast, is designed specifically to operate on sparse, non-sequential episodic transitions, allowing it to infer the structural relationships among states without needing long trajectories.

## B    EVIDENCE FOR BUILDING SPATIAL MAPS FROM DISJOINT EXPERIENCES IN ANIMALS

Several behavioral studies have demonstrated that animals and humans can integrate information collected across disjoint episodes into coherent spatial or relational maps. (Roberts et al., 2007) showed that rats inferred a shortcut between two locations in a maze after exploring its subcomponents in separate sessions on different days. From the perspective of our framework, the animals observed disconnected edges of the underlying spatial graph—never experiencing them as a continuous trajectory—yet were still able to infer the globally shortest route during the test. (Fernandez et al., 2023) similarly demonstrated that humans trained to navigate locally within three separate environments on different days could integrate these fragmented experiences to navigate across them globally. (Park et al., 2020) found that human subjects who learned each dimension of a 2D relational space in separate episodes nonetheless formed a joint map of the full space, again illustrating cross-episode integration.

There is also substantial neural evidence that hippocampal and amygdala circuits integrate memories according to shared structure rather than temporal contiguity. (Yokose et al., 2017) showed that amygdala-dependent memories from distinct tasks form overlapping engrams when they share a critical stimulus. (Schlichting et al., 2015) found that the anterior hippocampus integrates across paired associates (A→B, B→C) to support the inferred A→C relationship. (McKenzie et al., 2014) reported that hippocampal population codes become similar for objects that share spatial context, even when those objects were experienced in separate episodes. Related reviews (Morton et al., 2017; Brunec et al., 2020) synthesize a broad literature showing that the hippocampus organizes memories into integrated representations that reflect spatial, temporal, and conceptual structure—often beyond what was directly and continuously experienced.

Altogether, these findings provide strong evidence that biological systems do face and solve a form of the "disjoint memory" problem. They routinely combine episodic fragments acquired at different times, in different contexts, and along different task dimensions, to construct coherent internal maps that support flexible behavior.

## C    EXTENDED METHODS

### C.1    TRAINING DETAILS

#### C.1.1    OPEN ARENA

In the `Open Arena` environment, each integer state value is encoded as a 6-bit binary. Each bit is independently projected—via either the source-state or end-state linear projector—into a 128-dimensional embedding. The six embeddings are concatenated into a single 768-dimensional state vector. Actions are similarly mapped into 768-dimensional embeddings using a dedicated linear projector. The final transition embedding is then obtained by averaging the three 768-dimensional vectors corresponding to (1) the start-state, (2) the action, and (3) the end-state.

### C.1.2 RANDOM WALL

In the `Random Wall` environment, the same procedure applies, except that each state is represented by an integer and projected into a 1,024-dimensional embedding, yielding a 1,024-dimensional transition vector.

### C.1.3 MINIGRID

In the Minigrid environment, the states are $5 \times 5$ egocentric view, observing 25 locations all at once. Each location in the environment is assigned one of the colored objects (e.g, red floor) among a pre-defined set of 9 possible colored objects. Each of the objects among the 25 observed objects in a state is encoded into a 64 dimensional vector separately. The encodings of the 25 objects are then concatenated to produce a single 1600 dimensional representation for the state. Similar to `Open Arena` and `Random Wall`, the $s_s$, $a$, and $s_e$ embeddings are averaged to produce the final embedding for the transition. The prediction for the state is 25 independent 9-way classifications.

### C.1.4 PROCTHOR EXPERIMENTS

In ProcThor (Deitke et al., 2022), an agent can be placed anywhere in the environment to collect $224 \times 224 \times 3$ RGB observations (i.e. states; FOV=90). A transition consists of a start state $s_s$, a continuous action vector $[\Delta x, \Delta y, \Delta \theta]$ specifying how much to move right, forward, and rotate clockwise (negative values mean moving along the opposite axis) to arrive at the end state, and an end state $s_e$. To sample a transition, we first spawn the agent at a random, unobstructed location with a random rotation (start state). Then we randomly teleport the agent to a new location that is within the $3m \times 3m$ box of the start state, followed by a random rotation (end state). Lastly, we compute the egocentric action vector from the start state to the end state's poses.

The states are embedded by DINOV3 (Siméoni et al., 2025) into $W \times H \times D = 14 \times 14 \times 1024$ feature map, padded and pooled into $W \times H \times D = 4 \times 4 \times 1024$, then flattened into 16, 1024-dimensional tokens. The action is projected by a 1-layer MLP into a standalone 1024-dimensional token. The start state, end state, and action tokens are concatenated into a sequence of 33 tokens to represent a single transition. Suppose a memory bank has $N$ memories, then ESWM receives the concatenation of the memory bank and query, totaling $(N + 1) \times 33$ tokens, as input.

The action and end state of the query transition are masked with equal chances (the start state is left out for our preliminary experiment). For state prediction, the model is tasked to maximize the cosine similarity between its output and DINOV3's representation. For action prediction, the model performs categorical prediction over a discretized action space. To discretize action prediction, the $3m \times 3m$ movement box is discretized into grids, with each grid representing a class. ESWM needs to essentially predict how many steps to move right/left, forward/backward until the end state's location is reached, with the step size (or discretization granularity) determined so that there are 64 classes. The rotation is discretized by 10-degree intervals, resulting in 36 classes.

The concatenation of the memory bank and query is fed into a multi-layer transformer. After all transformer blocks, we linearly read out the action prediction from the action token, and directly match the state tokens with their target representations without any further transformation. We apply cross-entropy loss to action predictions and cosine distance loss to states.

**ProcThor Training Pipeline** To train ESWM in GPU-based environments like ProcThor, we devise a memory-efficient, fast, and scalable training pipeline that largely follows DD-PPO (Wijmans et al., 2020). The training is distributed over $N$ GPUs, with each GPU alternating between online data generation across $K$ scenes and training. In our preliminary experiment, we train ESWM with 4 A100 GPUs for 19 days over 80k iterations with an effective batch size of 144, using only 250 GB of memory.

### C.1.5 MODEL ARCHITECTURES.

- **Transformer:** 8 attention heads per layer; feed-forward hidden dimension of 2,048, unless otherwise specified.
- **LSTM:** hidden state size of 1,024.

- **Mamba:** model dimension of 768 in `Open Arena` and 1,024 in `Random Wall`.

For ProcThor experiment, we replaced the standard transformer block with the Alternating Attention block from VGGT (Wang et al., 2025). Each transformer block consists of a memory-wise attention layer, where the attention scope is limited to the tokens within a transition – consolidating the relation between the start and end state, and a global attention layer, where all tokens can attend to each other – enabling memory integration. We use 16 heads in our ProcThor runs.

### C.1.6 TRAINING HYPERPARAMETERS.

All models were trained on a single NVIDIA A100 GPU for 480,000 iterations using:

- Optimizer: AdamW with initial learning rate $1 \times 10^{-4}$ and cosine decay schedule.
- Dropout: 0.1.
- Batch size: 128 memory bank and query pairs.
- Training time: 19–28 hours, depending on model capacity.
- Memory usage: 2GB. The low memory consumption is attributed to the online generation of data.

### C.2 BUILDING RANDOM 2D GRID ENVIRONMENTS

To generate an input (memory bank, query) pair to ESWM, we first generate an environment $e \in \mathcal{E}$. Each $e$ is represented as a 5-tuple $(G, g, W, \psi, A)$, where:

- $G = (L, E)$: An undirected graph, with $L$ representing unique locations and $E$ representing undirected edges connecting adjacent locations. The graph $G$ adopts a fixed hexagonal structure across all environments. It models grid-like movement dynamics and serves as the canvas for creating complex environments.
- $g$: A filtering function that generates a subgraph $G_{\text{obs}} \subseteq G$, containing only observable locations $L_{\text{obs}} \subseteq L$ and transitions $E_{\text{obs}} \subseteq E$ (i.e. part of the environment that will be observable to the agent in that sample).
- $W$: A subset of contiguous locations in $L$, representing a wall that restrict agent's motion in the environment.
- $\psi$: A mapping from $L$ to integer observations, assigning each location to a unique observation from $S \subseteq \mathbb{Z}$. This allows random association of locations and observed values (or states) in each sample.
- $A$: Maps an undirected edge $e = (l_i, l_j) \in E$, which is a pair of locations, to an action-induced transition either starting from $l_i$ or $l_j$ with equal chances. e.g. $A(l_i, l_j)$ returns a 3-tuple transition $(l_i, \text{go West}, l_j)$ or $(l_j, \text{go East}, l_i)$ with equal probability.

In the `Open Arena` setting (Section 3.3), the entire environment is observable and there are no obstacles (i.e, $g$ is the identity function, $W = \emptyset$). In the `Random Wall` setting (Section 3.3), $g$ removes a random subset $V_{\text{unobs}} \subseteq V$ along with their attached edges, and $W \neq \emptyset$. The components $g$, $W$, and $\psi$ are randomly generated for each environment $e$, where applicable.

### C.3 GENERATING MEMORY BANKS IN 2D GRID ENVIRONMENTS

The minimal spanning episodic memories $M$ for an environment $e = (G, g, W, \psi, A)$ is generated as follows:

1. Apply the filtering function $g$ to $G$ to produce the partially observed subgraph $G_{\text{obs}} = (L_{\text{obs}}, E_{\text{obs}})$.
2. Run a minimal spanning tree (MST) algorithm on $G_{\text{obs}}$, with random weights assigned to edges to ensure diversity. Tree effectively models a set of fragmented transitions that do not naturally integrate into a continuous trajectory. This produces a set of undirected edges $T = \{(l_i, l_j) \mid l_i, l_j \in L_{\text{obs}}\}$.

3. Use $A$ to map edges in $T$ to directed 3-tuple transitions between locations and $\psi$ to map locations to observations. This produces an array of transitions between observations $[(s_\text{s}, a, s_\text{e})]$.

4. Modify transitions ending at locations in $W$ to form self-loops, effectively modeling blocked movement. Then, randomly permute the array of transitions to produce the final array of transitions $M$.

In `Open Arena` and `Random Wall`, the memory banks $M$ observe at most 18 and 36 transitions out of the total 84 and 180 transitions, respectively.

### C.4  QUERY SELECTION

The resulting memory bank $M$ is an unordered set of one-step transitional episodic memories that span all observable locations in the environment (Fig. 2). Given $M$, the model is tasked to infer the missing component of a query transition $q = (s_\text{s}, a, s_\text{e})$.

For `Random Wall`, the query $q$ is selected from 1) the set of transitions that are in $G_\text{obs}$ but not in the memory bank (unseen transitions; 68%); 2) from the memory bank (seen transitions; 15%) and; 3) from a set of unsolvable transitions where one end is in $L_\text{unobs}$ (17%). In `Open Arena`, the query $q$ is always selected from unseen transitions.

The model receives the $M$ and $q = (s_\text{s}, a, s_\text{e})$, with either $s_\text{s}$, $a$, or $s_\text{e}$ masked randomly with equal probability, as input, and asked to infer the masked component.

### C.5  TEM-T IMPLEMENTATION

Our goal is to maximally equalize the training of TEM-T and ESWM to eliminate potential confounds such as task differences. Thus, TEM-T also receive the concatenation of memory bank and unseen query transition as input, except that the disjoint memory bank is replaced with a sequential trajectory that spans the entire environment. The task is to predict the missing end-state of the final query transition, analogous to inferring the masked end state in ESWM. The trajectory is kept minimal, generated by `NetworkX traveling salesman problem`, to match the sparsity of the memories we provide for ESWM. In `Random Wall`, we explicitly ensure that the agent runs into each obstacle at least once throughout the traversal so that the model has sufficient knowledge to infer the underlying structure of the environment.

To our best knowledge, no publicly available code exists for TEM-T. Therefore, we resort to a custom implementation that closely follows the authors' recipe. We use the same dimensions to encode the positions $e_t$ and observations $x_t$ (384 in `Open Arena` and 512 in `Random Wall`) and ensure the combined dimension of $h_t = [x_t, e_t]$ matches ESWM's embedding dimension. Sigmoid is used as the non-linear activation $\sigma$ in positional encoding generation, following the observation that linear and ReLU activation produce exploding gradients. We use 8 heads and 2048 feed-forward dimensions, matching the configuration of ESWM. The prediction is read out linearly from the embedded query transition after a single/multiple transformer blocks. We train TEM-T over 460k iterations with a batch size of 128 and a cosine learning rate schedule, same as ESWM.

### C.6  ISOMAP

We consider ESWM's population neuron activity at a particular location as the ESWM's activation vector as it infers the missing component of a query transition that either starts or ends at that location. For $s_e$ and $a$ predictions (i.e, $s_s$ is visible to ESWM), the activation vector is associated with the $s_s$'s location, while for $s_s$ prediction (i.e, $s_s$ is not visible), we associate the activation with the $s_e$'s location. For transformer models, we used the query token after each transformer encoder block as the activation vector, while for multilayer LSTM and Mamba, we used the final hidden state (i.e, the last token from the processed sequence) from the last layer.

To collect ESWM's activation across all locations in an environment, we task ESWM to do either $s_s$ $a$, or $s_e$ prediction for all transitions in the environment conditioned on a memory bank. This process is repeated with multiple memory banks from the same environment until approximately

1,000 activation vectors are collected. We then apply ISOMAP using cosine distance and 20–60 neighbors to generate the visualizations.

## C.7 EUCLIDEAN DISTANCE ESTIMATION FROM ACTIVATION

A linear layer receives as input the concatenation of three vectors: the activations from Transformer-2L ESWM corresponding to action predictions starting from states A, B, and C. It outputs a scalar representing the probability that the Euclidean distance between A and B exceeds that between A and C. The classifier is trained on 3,000 state triplets sampled all from different environments and evaluated on 1,000 held-out triplets. Reported accuracies are averaged over 10 runs with different randomly initialized weights for the linear layer.

## C.8 sRSA

We use the formula from (Levenstein et al., 2024) to compute sRSA. This metric measures the Spearman correlation between latent and physical distances. To get the latent representation at a particular location, we task the model to solve all queries that start in that location and average the activation. We repeat this procedure across all locations in an environment and across 5 memory banks from each environment. We used ESWM-T-2L's first layer's query token's activation vector during the action prediction task for this analysis.

## C.9 NOISY MEMORY BANKS IMPLEMENTION

To create memory banks with noisy, or contradictory, state observations, we pick a location and change the corresponding memories in the memory bank to observe different states at that location.

To create memory banks with noisy, or contradictory, transitions, we randomly pick a transition from the memory bank, edit its end state, and append the modified transition to the memory bank.

This analysis is done with ESWM-T-4L in `Random Wall`.

## C.10 BASELINE NAVIGATIONAL AGENT

The baseline agent is trained via A2C (Sutton, 2018; Mnih, 2016) to navigate between any two states in a single `Random Wall` environment, which differs from ESWM and EPN, which are meta-trained across many environments. At each time step $t$, the agent receives as input a sequence of historical interactions with the environment, each as a tuple $(s_{t-1}, a_{t-1}, s_t, s_{goal})$, and is tasked to output an action towards the goal state $s_{goal}$. Each part of the tuple is distinctly embedded onto a 128D vector and the vectors are averaged to produce the final observation at each time step. The embedded sequence is fed into a one-layer LSTM (with a hidden dimension of 256) and the predicted action and value are read out from the final hidden state via distinct linear heads.

The agent receives a penalty of -0.01 every time step except when it reaches the goal state, where it receives a reward of 1. The training is done over 5000 episodes with a discount factor of 0.99 and learning rate of $1 \times 10^{-3}$.

# D  ADDITIONAL ANALYSIS

## D.1  COMPARING MODELS IN OPEN ARENA

Different models exhibit drastically different performance in Open Arena. This section aims to provide insights into the factors that contribute to each model's performance.

The ESWM-T models perform noticeably well in the Open Arena. We investigate which hyperparameter affect its performance the most. In Fig.2, we fix the number of heads and observe a positive correlation between model depth and performance. In Table. 11, we fix the depth and vary the number of heads. We observe that while increasing depth leads to a monotonic increase in performance, increasing the number of heads initially improves performance, but exceeding 8 heads leads to a quick drop in performance. In addition, the model depth impacts all three prediction tasks, while the number of heads selectively impacts state prediction tasks more than the action prediction task.

As shown in Fig.8, ESWM-MAMBA and ESWM-LSTM both overfit on the training states, explaining their poor performance in completely unseen test states. However, we note an additional observation: LSTM struggles to perform even on the training set in Open Arena, unlike in Random Wall where it succeeds on both train and evaluation splits. We suspect another task difference between Open Arena and Random Wall contributes to this discrepancy: states in Open Arena consist of a 6-bit value, each represented by a 128-dimensional vector while states in Random Wall consist of a single continuous vector. The latter may allow a more compact latent representation that better fits within LSTM's limited memory capacity.

TEM-T models, which also incorporate transformer into its architecture, underperform compared with ESWM-T models. TEM-T is architecturally distinct (See Section 2) and is trained on a different distribution (minimal spanning trajectories rather than minimal spanning trees) compared to ESWM-T. To determine which of these two factors contributes to the performance gap, we train ESWM-T-10L on TEM-T's training distribution and find that it can reach a similar performance compared with training on ESWM's, predicting unseen $s_s$, $a$, $s_e$ with accuracies 99%, 99%, 99%. The similar is true in Random Wall, where ESWM-T-6L reaches $s_s$, $a$, $s_e$ accuracies of 81%, 89%, 78% when trained with TEM-T's training distribution. These results indicate that TEM-T's architecture most likely explains its performance gap with ESWM-T.

## D.2  COMPARING ESWM TRAINED WITH OR WITHOUT THE MINIMALITY CONSTRAINT ON MEMORY BANKS

We show that ESWM-T performs well when trained with minimal memory banks (Fig. 2) and non-minimal memory banks (Fig. 9). However, ESWM-T trained on non-minimal memory banks, where more environmental transitions are observed, underpeforms the one trained on minimal memory bank, where less environmental transitions are observed.

To investigate why the ESWM-T trained on non-minimal memory banks underperforms while having access to more experience in an environment, we break down the model's performance across memory bank sizes and observe a substantial gap between denser and sparser memory banks, even though they are equally likely to be in the training data (Fig. 9). This suggests that, when trained on a mixture of densities, the model focuses more on solving the easier tasks where it is given an over-complete memory bank, and only partially succeeds in learning the more challenging tasks when minimal information is given. This leads to weaker spatial reasoning skills and poorer performance when the observation pattern changes.

In contrast, the ESWM-T that is trained under minimality constraints is forced to solve the harder spatial reasoning problem consistently. As shown in the (Fig. 4c) this produces representations that transfer robustly to denser, Out-of-Distribution (OOD) memory banks. We also note that within the training distribution (ID), increasing memory bank size corresponds to an expansion of the observed area, which requires the model to coherently build a larger map which is more challenging and likely explains the performance drop.

Collectively, these findings suggest that applying sparsity constraints during training leads to better test-time flexibility in terms of information availability.

### D.3 HANDLING DISCONNECTED MEMORY CLUSTERS

During `Random Wall` training, memory banks can contain disconnected memory clusters, with no paths between observations in distinct clusters. This leads to uncertainty in spatial relationships as path integration provides no avail. In rare cases, the model is asked to solve queries that bridge disconnected clusters. This raises two questions: (1) Does the model infer the relative positions of disconnected memory clusters (e.g., which part of the room each cluster maps onto)? (2) Which cues enable these inferences?

#### D.3.1 INFERENCE FROM ENVIRONMENTAL STRUCTURE

We observe that when memory clusters are disconnected, the model leverages the underlying structure of the room to infer their relative positions. For example, the model aligns memory clusters with the room's boundary when their shape matches. This allows the model to piece together disconnected memory clusters, like solving a puzzle, by identifying the configuration that satisfies the invariant structural constraints. Evidence for this behaviour is twofold. First, in the model's latent space, the disconnected clusters are stitched together, rather than overlapping or disconnecting, to form a smooth surface reflecting the room's structure (Fig. 3c). Second, the model's end state prediction distributions, starting from the boundary states of one cluster, are biased towards the boundary states of the inferred neighbouring cluster, instead of exclusively outputting the "I don't know" option.

#### D.3.2 GENERALIZATION FROM OBSERVATIONS OF OBSTACLES

In cases where disconnected memory clusters' shapes do not provide clear cues about their relative locations, the model relies on observations of obstacles to deduce their positions. For instance, given two disconnected clusters of memories, the model initially maps them to different locations in its latent space and outputs predictions exclusively reflecting uncertainty ("I don't know"). However, if both clusters observe obstacles, even though the clusters remain disconnected, the model applies its knowledge of common wall shapes (e.g, walls are usually straight) to make inferences about the relative locations of the clusters. In the latent space, the two previously disconnected clusters are connected and aligned along the inferred wall shape to form a continuous representation (Fig. 3d, 13). The model's predictions are also biased towards the inferred neighbours. Moreover, a single observation of obstacles from both clusters is sufficient for the model to infer the shape of the wall, and therefore infer the relative positions of the clusters (Fig. 13). These results demonstrate that ESWM can leverage the invariant structure of the room and combine observations of obstacles with prior knowledge of wall shapes to construct a coherent spatial representation.

### D.4 PLANNING BY GUIDED IMAGINATION

While the strategy introduced in the main paper enables planning, it uniformly searches through the space, unbiasedly imagining the consequences of all actions, even if some move further from the goal. In contrast, animals with internal maps encoding geometric constraints plan biasedly, prioritizing actions that shorten the geodesic distance (i.e, the shortest path length) to the goal. We show that the spatial maps in ESWM's latent space, which accurately encode obstacles and geodesic distance between states (Fig. 16b), can be used for guided imagination towards the goal.

The previous strategy is akin to Dijkstra's algorithm, or A* with a null heuristic function (Hart et al., 1968; Ferguson et al., 2005). The key to efficient planning, which prunes undesirable actions early, then, is a heuristic function containing rich information on the geodesic distances between states in the environment. Inspired by ISOMAP Tenenbaum et al. (2000), we construct such a heuristic function $h_{latent}$ as follows (See Alg.4 for pseudocode): 1) Query the model to solve the end state prediction task for all possible $(s, a)$ pairs, collecting activations. 2) Construct a graph $G_{latent}$ where nodes are activations and edges connect nodes within radius $R_{latent}$ (Appendix G). Edges are weighted by the cosine distance between their two endpoints (Fig. 16b). Intuitively, each node on $G_{latent}$ is the activation of predicting the outcome of action $a$ starting at state $s$. 3) Compute pairwise shortest paths on this graph to create a geodesic distance table $h_{latent}$.

$h_{latent}$ captures ESWM's latent spatial map into a look-up table containing information on which action takes you closer to the goal state at every step of imagination. A* guided imagination with

$h_{\text{latent}}$, improves planning efficiency by 57% (Fig. 16a), as measured by the number of states visited, over Dijkstra's while maintaining path optimality (Fig. 16c). Furthermore, greedy policy based on $h_{\text{latent}}$ achieves 70% success in navigating around obstacles (Fig. 16a). These results suggest that structured latent spaces in world models can serve as efficient heuristics for planning.

# E ADDITIONAL TABLES

|  | 2 Heads | 4 Heads | 8 Heads | 16 Heads |
|---|---|---|---|---|
| Predict Source | 40.4 | 78.9 | **83.3** | 42.0 |
| Predict Action | 84.6 | 91.8 | **92.6** | 91.8 |
| Predict End | 37.6 | 78.1 | **79.6** | 40.4 |

Table 1: **ESWM-T number of heads ablation**
. We report the accuracies for the start state, action, and end state prediction task in `Open Arena` with ESWM-T-6L.

|  | Predict Source | Predict Action | Predict End |
|---|---|---|---|
| Seen | 0.9050 | 0.8700 | 1.0000 |
| Unseen | 0.8750 | 0.8880 | 0.9030 |
| Unsolvable | 0.8270 | 0.9050 | 0.8180 |

Table 2: **ESWM-T-4L performance in `Random Wall`**
. Accuracies for source, action, and end prediction.

|  | Predict Source | Predict Action | Predict End |
|---|---|---|---|
| Seen | 0.8780 | 0.8690 | 1.0000 |
| Unseen | 0.6660 | 0.7530 | 0.7030 |
| Unsolvable | 0.7620 | 0.8830 | 0.7560 |

Table 3: **ESWM-LSTM-4L performance in `Random Wall`**
. Accuracies for source, action, and end prediction.

# F ADDITIONAL FIGURES

|            | Predict Source | Predict Action | Predict End |
|------------|----------------|----------------|-------------|
| Seen       | 0.9000         | 0.8810         | 1.0000      |
| Unseen     | 0.7710         | 0.8110         | 0.8230      |
| Unsolvable | 0.8110         | 0.8990         | 0.8010      |

Table 4: **ESWM-Mamba-5L performance in `Random Wall`**. Accuracies for source, action, and end prediction.

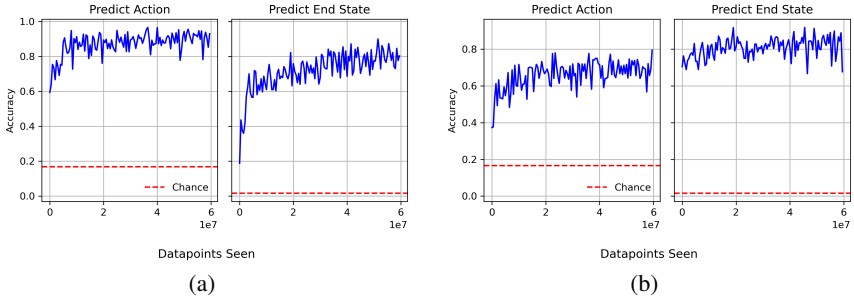

Figure 7: **ESWM's evaluation accuracy in grid environments with duplicated states.** A trajectory is prepended to the query transition to serve as context for disambiguating two locations with the same state. The context trajectory begins at a random state and ends at $s_s$ of the query transition. The query transition's $a$ and $s_e$ are masked with equal chances. **a)** Accuracy in `Open Arena` for ESWM-T-8L and **b)** `Random Wall` for ESWM-T-4L.

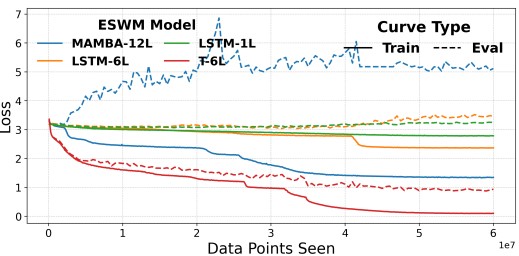

Figure 8: **Training and evaluation loss for various ESWM models in Open Arena.**

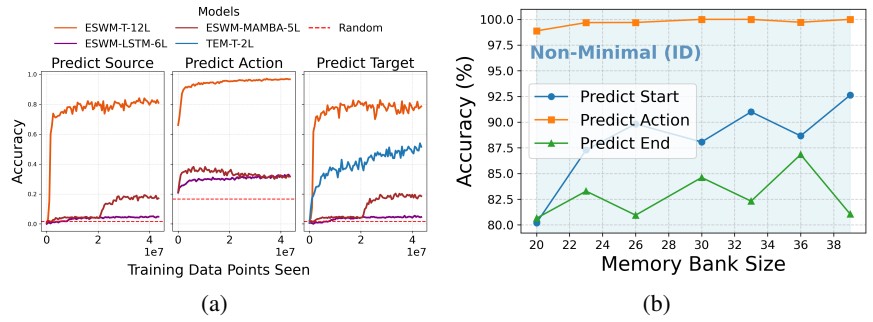

Figure 9: **Training with non-minimal memory banks.**
**a)** Evaluation accuracy on unseen queries for various models trained with non-minimal memory banks. ESWM receives memories that are *disjoint*, *spanning*, but *not minimal* (constructed by adding extra episodic memories to the minimal memory banks) and is tasked to predict an unseen query. TEM receives a minimal spanning trajectory padded with random walks as input. **b)** Prediction accuracy over memory bank sizes for ESWM-T-12L from **a)**. The shaded area emphasizes that this model is trained on non-minimal memory banks. ID means In Distribution.

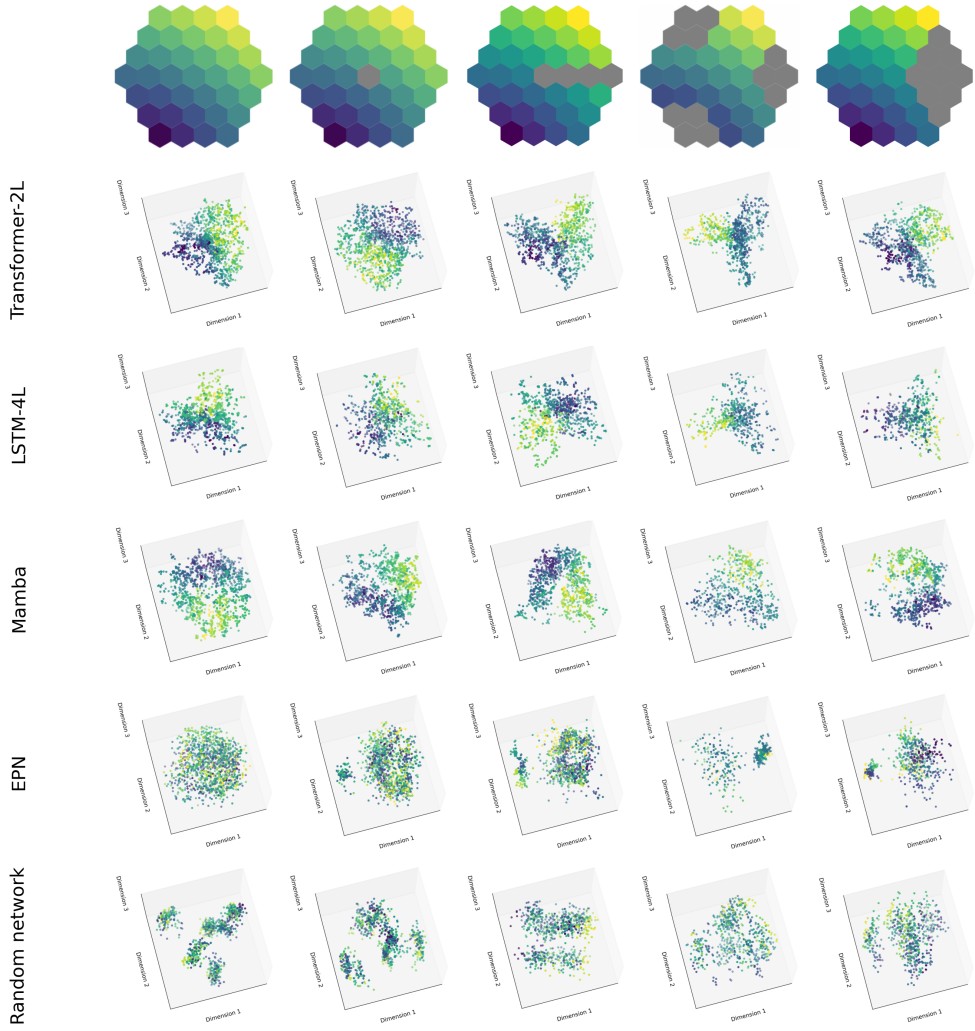

Figure 10: **ISOMAP projection of various ESWM-trained models' latent space.** The last two environments are **out-of-distribution** — ESWM has never seen multiple walls in the same room nor a cluster of obstacles that does not form a long, straight wall during training. For Transformer-2L, LSTM-4L, and Random network (LSTM-4L initialized with random weights), the activations for end-state prediction are used while for Mamba, the activations for action prediction are used. EPN (Section H) is an episodic-memory-based navigational and exploration agent trained on the same environment as ESWM.

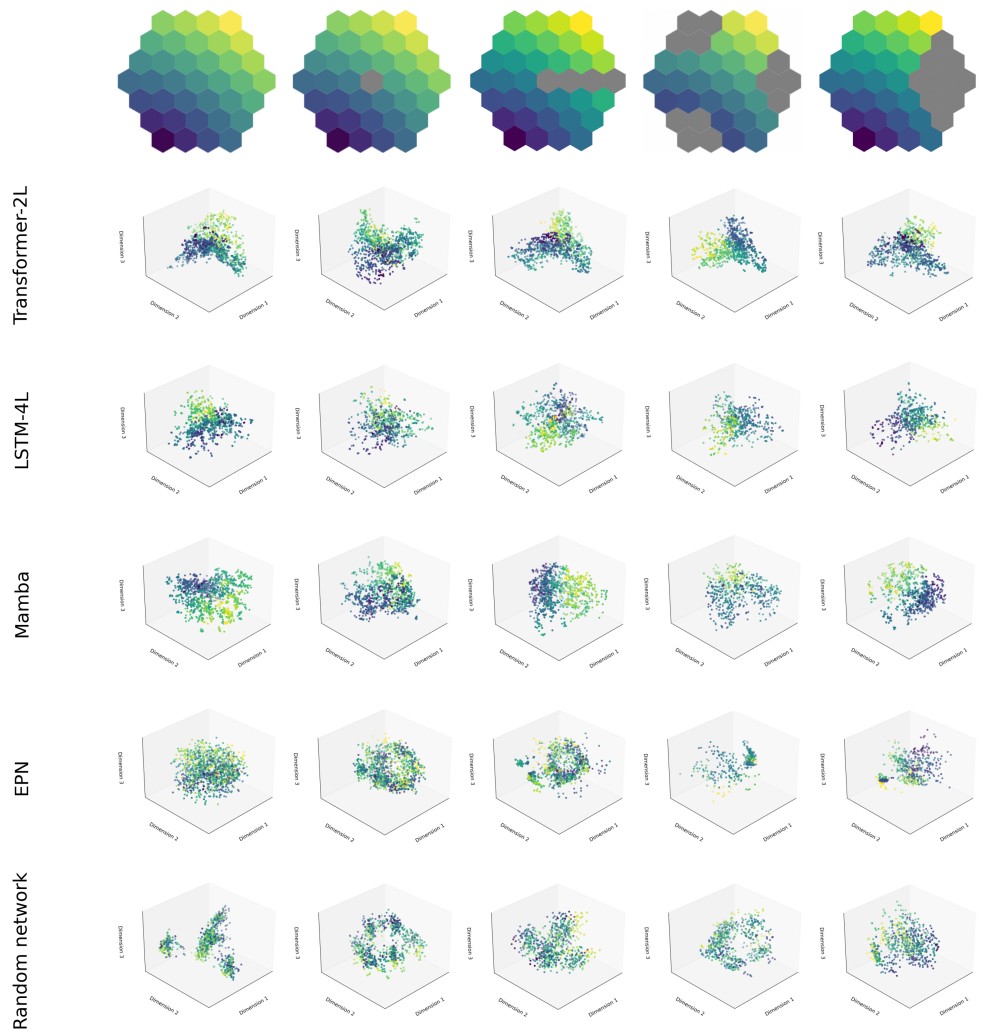

Figure 11: Same ISOMAP projections as Fig. 10 but in a different angle.

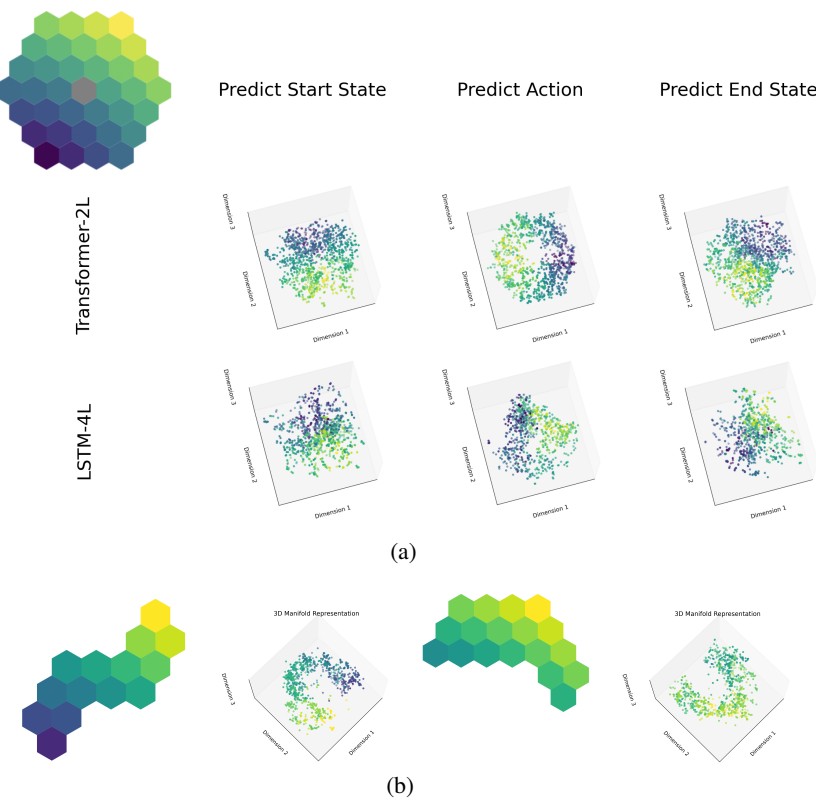

Figure 12: **ESWM's latent spatial map persists for all prediction tasks and partial observations.**
**a)** ESWM's latent spatial map for $s_s$, $a$, and $s_e$ predictions. Mamba is excluded as the spatial map is only found for the action prediction task. For Transformer-2L, the spatial map is found in the first layer for $a$ prediction, and the second layer for $s_s$ and $s_e$ prediction. **b)** Spatial map when the memory bank observes an irregular subsection of the room. Transformer-2L's first layer's activation for action prediction is used.

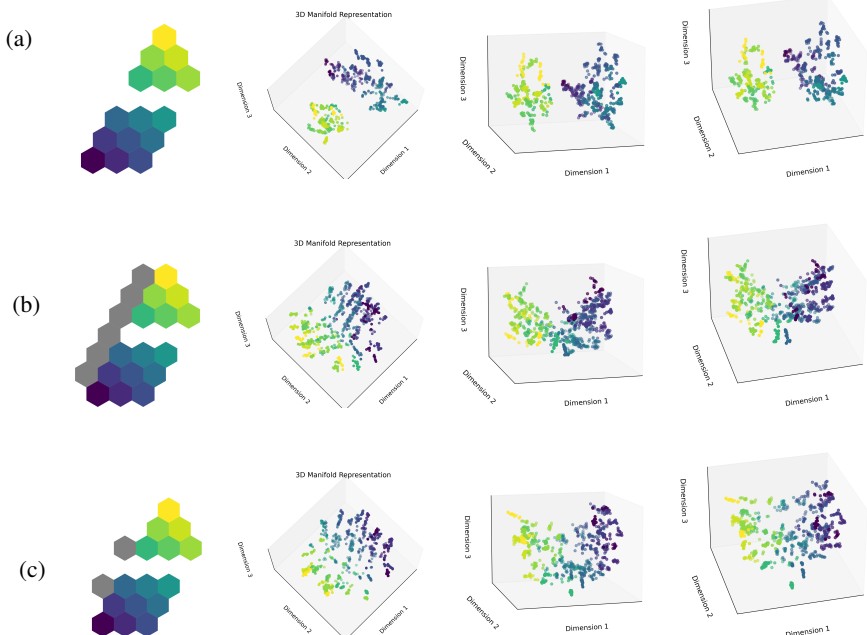

Figure 13: **ESWM's latent spatial map adjusts to structural changes.** ISOMAP projections of the 14-layer transformer model's seventh layer's activations. **a)** ESWM's latent space when the input memory bank $M$ consists of two disjoint memory clusters. **b)** ESWM's latent space when both memory clusters observe a wall to their left; despite remaining separate in the memory bank, ESWM inferred them to be connected as reflected in its latent space. **c)** ESWM's latent space when both memory clusters observe a single obstacle to their left. ESWM, again, inferred the clusters are connected but with sparser observations of obstacles than in b). 900 activations and 45 neighbours are used to fit the ISOMAP.

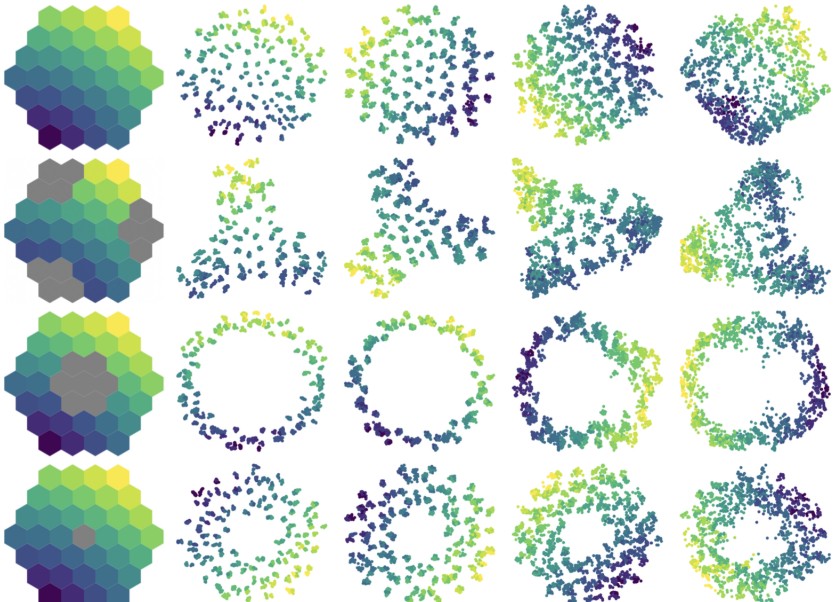

Figure 14: **Representation becomes more continuous with depth.** Left to right are 30-neighbours ISOMAP projections from layer 1, 6, 9, and 14 of Transformer-14L (2000 activations). The model was trained without random masking of subregions. Representation in the latent space becomes progressively more continuous.

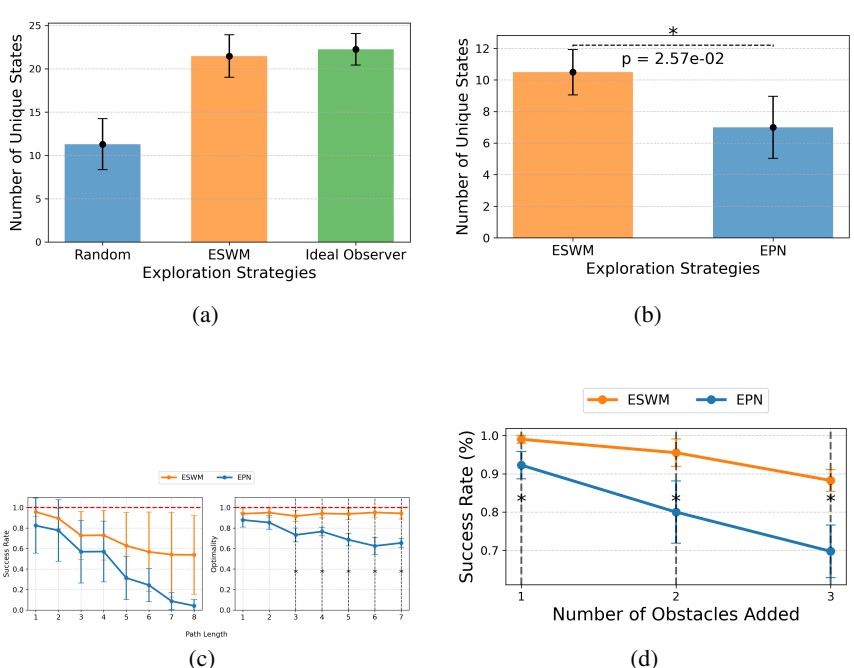

(a)

(b)

(c)

(d)

Figure 15: **Exploration, navigation, and adaptability for ESWM and EPN. a)** Number of unique states visited by different exploration strategies in a Random Wall environment with 37 locations over 25 time steps (n=1000). On average, ESWM explores a comparable number of states to an oracle explorer with access to obstacle locations. **b, c, d)** Same as Fig. 5 except the performance of ESWM and EPN is averaged over 4 random seeds. Error bars show standard deviation across seeds. A Welch's two-sample t-test was used to compare ESWM and EPN ($\alpha = 0.05$). A dashed line with "*" indicates a statistically significant difference ($p < 0.05$)

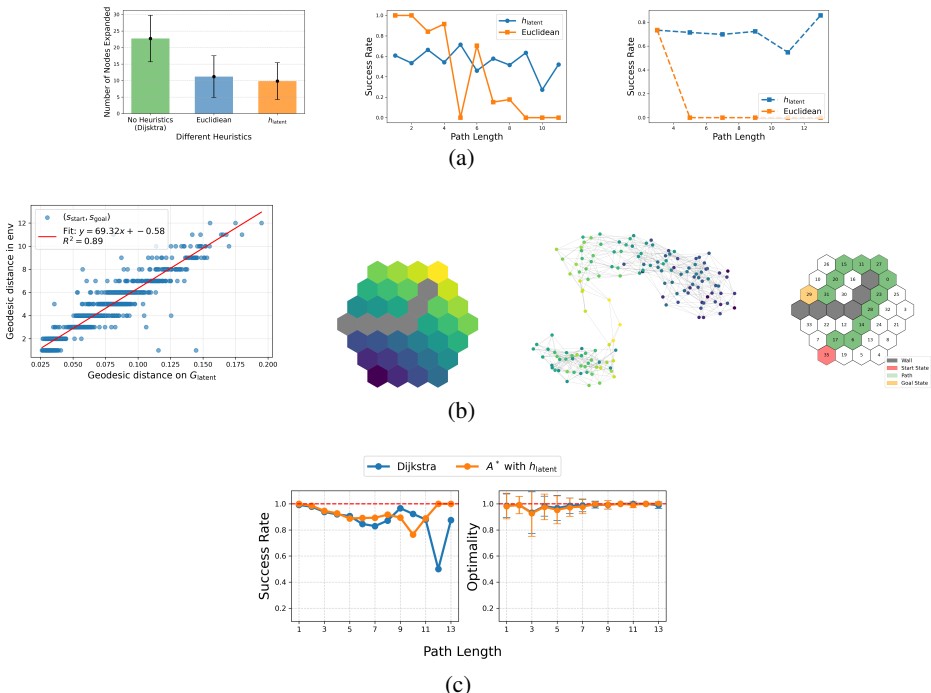

Figure 16: **The latent space of ESWM can be harnessed as a heuristic for more efficient planning**. All the results are on Random Wall environment with 37 locations. **a)** (Left) Average number of nodes expanded for different heuristics over 1000 paths. (Middle) Success rates of greedy navigation using different heuristics over 2400 paths. (Right) Success rates for greedy navigation on paths requiring detours around obstacles over 800 paths. **b)** (Left) Correlation between geodesic distances on $G_{\text{latent}}$ (see D.4) and corresponding distances in the environment, using n=1500 pairs of states $(s_{\text{start}}, s_{\text{goal}})$ over 75 environments. The linear fit shows a strong positive correlation with $R^2 = 0.89$. (Mid-Left) sample grid environment. (Mid-Right) $G_{\text{latent}}$ for the same environment. (Right) Agent navigates greedily on $h_{\text{latent}}$ on a path that requires detours around obstacles. **c)** Comparison of navigation success rate and path optimality between Dijkstra's and A* over different path lengths (n=1500). Although not guaranteed to be admissible, $h_{\text{latent}}$ empirically preserves path optimality.

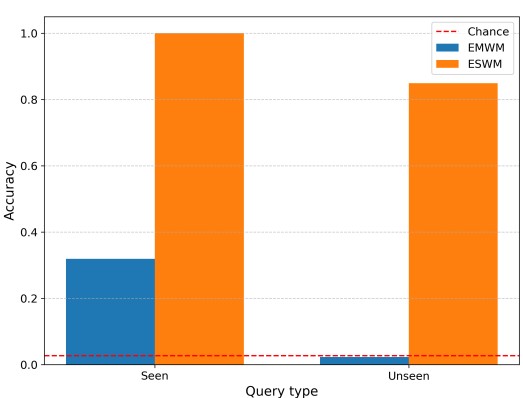

Figure 17: **Comparison of end-state prediction accuracy between EMWM Coda-Forno et al. (2022a) and Transformer-2L ESWM in `RandomWall`**. EMWM predicts using a weighted average over the top-K matches (K=5), but we found K=1 performs best, likely due to the sparsity of our memory banks (i.e, transitions are unique, so averaging more than one transition offers no benefit). To adapt the author's implementation to our task, we embed each token (transitional memory tokens or the end-state masked query) by averaging the embedding of each item $(s_t, a_t, s_{t+1})$, each produced using a distinct MLP layer – identical to ESWM. All tokens are sequentially fed into a 1-layer GRU Cho et al. (2014) to obtain the sequence of hidden states, which become the keys with the corresponding end-state embeddings as values for the memory table. The network is optimized to predict the $s_{t+1}$ for each token.

## G  ALGORITHMS

---

**Algorithm 1** Dijkstra-Style Pathfinding with ESWM

---

1: **procedure** FINDPATH(ESWM, memoryBank, $s_{\text{start}}$, $s_{\text{goal}}$, $T_{\max}$)
2:     **const**   $ACTION\_COST$
3:     Initialize priority queue $OpenSet$
4:     Initialize dictionary $Cost$
5:     Initialize dictionary $Parent$
6:     $Cost[s_{\text{start}}] = 0$, $Parent[s_{\text{start}}] = nil$
7:     Push $(s_{\text{start}}, cost = 0, depth = 0)$ into $OpenSet$
8:     **while** $OpenSet \neq \varnothing$ **do**
9:         $(u, c, d) \leftarrow$ PopMinCostNode($OpenSet$)
10:         **if** $u = s_{\text{goal}}$ **then**
11:             **return** ReconstructPath($Parent$, $u$)
12:         **end if**
13:         **if** $d \geq T_{\max}$ **then**
14:             **continue**                         $\triangleright$ remove path that exceeds horizon
15:         **end if**
16:         **for all** action $a$ in PossibleActions **do**
17:             $v \leftarrow$ PredictNextState($ESWM, memoryBank, u, a$)
18:             $c' \leftarrow c + ACTION\_COST$
19:             $d' \leftarrow d + 1$
20:             **if** $v$ not in $Cost$ **or** $c' < Cost[v]$ **then**
21:                 Push $(v, c', d')$ into $OpenSet$
22:                 $Cost[v] = c'$
23:                 $Parent[v] = u$
24:             **end if**
25:         **end for**
26:     **end while**
27:     **return** FAIL()                             $\triangleright$ no path within horizon $T_{\max}$
28: **end procedure**

---

---

**Algorithm 2** Explore Function

---

1: **procedure** EXPLORE(ESWM, memoryBank, $s$, actions)         $\triangleright$ $s$ is current observation
2:     Initialize $q$ as a max heap
3:     **for all** $a \in$ actions **do**
4:         $s', logits \leftarrow$ ESWM.PREDICTEND($memoryBank, s, a$)
5:         $probs \leftarrow$ SOFTMAX($logits$)
6:         **if** $s' =$ 'I don't know' **then**
7:             Enqueue $a$ into $q$ with cost $= 1 - \beta$ENTROPY($probs$) $\triangleright$ $\beta = 0.2$ in our experiments
8:         **else if** $probs[s'] \leq 0.8$ **then**
9:             Enqueue $a$ into $q$ with cost $= \gamma$ENTROPY($probs$)       $\triangleright$ $\gamma = 0.1$ in our experiments
10:         **end if**
11:     **end for**
12:     **return** $q$.REMOVEMAX()
13: **end procedure**

---

---

**Algorithm 3** Get all State-Action Pairs

---

 1: **procedure** GETSAS(memoryBank)
 2:     $S \leftarrow$ GETUNIQUESTATES(memoryBank)
 3:     Initialize $l$ as an empty list
 4:     **for all** $s \in S$ **do**
 5:         **for all** $a \in$ Actions **do**
 6:             $l$.append($(s, a)$)
 7:         **end for**
 8:     **end for**
 9:     **return** $l$
10: **end procedure**

---

---

**Algorithm 4** Compute Geodesic Distance Table $h_{\text{latent}}$

---

 1: **procedure** GETHEURISTICS(ESWM, memoryBank, $R_{\text{latent}}$)
 2:     $SAs \leftarrow$ GETSAS(memoryBank)
 3:     $N \leftarrow |SAs|$
 4:     Initialize $nodesToSAs$ as an empty dictionary
 5:     Initialize $activations$ as an empty list
 6:     Initialize $h_{latent}$ as a matrix of shape $N \times N$
 7:     **for all** $(s, a) \in saPairs$ **do**
 8:         $activation \leftarrow$ ESWM's activation for end state prediction task starting at $s$ taking action $a$
 9:         $nodesToSAs[activation] \leftarrow (s, a)$
10:         $activations.append(activation)$
11:     **end for**
12:     $G_{\text{latent}} \leftarrow$ NEARESTNEIGHBOURGRAPH($activations$, $R_{\text{latent}}$)
13:     **for all** $n_i \in G_{\text{latent}}.nodes$ **do**
14:         **for all** $n_j \in G_{\text{latent}}.nodes$ **do**               ▷ $n_i, n_j$ are activations
15:             $sa_i, sa_j \leftarrow nodesToSAs[n_i], nodesToSAs[n_j]$
16:             $h_{latent}[sa_i, sa_j] \leftarrow$ SHORTESTPATHLENGTH($G_{\text{latent}}, n_i, n_j$)
17:         **end for**
18:     **end for**
19:     **return** $h_{latent}$
20: **end procedure**

---

Note that $h_{\text{latent}}$ approximates the geodesic distance between any two state-action pairs rather than between any two states. Geodesic distances between states can be obtained by averaging over the action space.

**Selection of $R_{\text{latent}}$**

In essence, $R_{\text{latent}}$ is the radius among a set of radii R in which the corresponding $h_{\text{latent}}$, obtained from Alg.4, best captures the geodesic distance information in the environment. More specifically, $R_{\text{latent}}$ is the $r \in R$ that maximizes both the success rate and the optimality of the agent's path when it navigates greedily to the corresponding $h_{\text{latent}}$. We consider a navigation as successful if the the agent reaches the goal state within 20 time steps. Path optimality is measured by the ratio between the shortest path and the agent's path. To determine $R_{\text{latent}}$, we conducted a search over a bounded range $R$, which includes those whose ISOMAP projections accurately reflect the structure of the environment.

**Limitations of Alg4**

One drawback is that $h_{\text{latent}}$ can overestimate the true geodesic distance, i.e. it is non-admissible. This is because $G_{\text{latent}}$ might be under-connected for the given $R_{\text{latent}}$ and memory bank (i.e. nodes aren't connected by an edge when they should). This can be mitigated by using a larger radius $R_{\text{latent}} + \epsilon$ to build $G_{\text{latent}}$ (i.e. prefer over-connection over under-connection). However, we acknowledge that there is no way to guarantee admissibility, hence, the optimality of the path. This

is a trade-off for faster search. In future work, we will explore adaptive radius where radius, hence $h_{\text{latent}}$, can be adjusted dynamically as agents interact with the environment.

## H  EPISODIC PLANNING NETWORKS (EPN)

The Episodic Planning Networks (EPN) model Ritter et al. (2020) is specifically designed and trained to rapidly learn to navigate in novel environments. Comparatively, ESWM is a general-purpose model not explicitly trained for navigation. By comparing with EPN, we investigate whether ESWM's ability to construct internal models of spatial structures enables navigational capabilities. Similarly to ESWM, EPN is designed to leverage episodic memory to facilitate rapid adaptation in unfamiliar environments, and both models are episodic memory-driven transformer-based models.

**Hyperparameters** – We use the same hyperparameters as those reported by Ritter et al. (2020).

**Training** – Both models are trained on the same environment structure variations and amount of data, approximately $1.14e9$ steps, which is a restrictive data regime for EPN. EPN is trained to navigate through reinforcement learning, using IMPALA (Espeholt et al., 2018). ESWM is trained to build internal model of environments from sparse disjointed episodic experiences through self-supervision.

**Memory bank** – EPN has a memory bank containing 200 transitions, whereas ESWM only stores the minimal spanning tree, with 19 transitions for smaller environments and 36 for larger ones. When evaluating EPN ability to navigate, we follow the methodology from Ritter et al. (2020). EPN explores to populate its memory bank for the first 2/3 of episodes , and the performance is only measured on the last third.

## I  IMPACT STATEMENTS

This paper aims to advance the foundations of machine learning by developing a model for structured memory and spatial understanding. Although our primary contributions are theoretical and methodological, the ideas explored here connect directly to real-world applications in autonomous exploration, navigation, and decision-making. In particular, the compact latent representations and rollout capabilities of ESWM make it a promising component for future embodied systems operating in unstructured physical environments.

Potential positive societal impacts include safer and more efficient navigation technologies, improved exploration strategies for robotics, and progress toward AI systems that reason about space in ways inspired by human cognition. At the same time, we recognize the risks associated with deploying autonomous agents in real-world settings, including the possibility of collecting sensitive data or influencing high-stakes decisions. These considerations underscore the importance of responsible development, careful evaluation in physical contexts, and adherence to ethical guidelines when integrating such models into embodied platforms.

