# OpenReview forum: "Building spatial world models from sparse transitional episodic memories"
_ICLR.cc/2026/Conference — ICLR 2026 Poster_

### Official Review · Reviewer_o4ni · 2025-10-22

**Soundness:** 3
**Presentation:** 4
**Contribution:** 3
**Rating:** 8
**Confidence:** 3

**Summary:**

Paper proposes a novel representation of spatial memory for mapping and navigation. A transformer-based model is trained via mask-prediction self-supervised method to predict the unseen region/path. Visualization of the neural network with ISOMAP projection shows that it is able to learn to infer the relative position of disjoint regions. The framework is further extended to the exploration task and goal-seeking task. It can also effectively handle changing environment for the goal-seeking task.

**Strengths:**

The inspiration from neuroscience has been a common thread in the major breakthroughs in computational science. In particular, the neural network is proven to be instructional in how computational theories are advanced through such inspiration. While this paper does not directly model brain episodic memory, the concept being borrowed/inspired shows good potential in the application of mapping and navigation.

One of the major challenges facing machine learning and AI is the effective and efficient representation and modelling of memory within/coupled with neural network training. This work may lead to some interesting ideas for other researchers. S1. Claims are supported by the strong experimental results for the exploration tasks and the goal-seeking task (without global map).

S2. Proposed methods are sensible and applicable for the problem.

S3. Experimental designs and analyses are sound and valid. There are sufficient experiments and ablation studies to support the central claim of the paper. There are additional experiments to compare transformer variants of the model, ESWM-T, against transformer-based such as TEM-T.

**Weaknesses:**

Minor: The proposed solution may not generalized to continuous environments. However, given the highly theorical framing of the paper, this is not a major concern.

**Questions:**

1. Does the authors have specific plans for generalizing their methods for real-world environment, i.e. physical embodiment, for their future work?

**Details Of Ethics Concerns:**

No concerns as all dataset/benchmarks are synthetic and without any personal data etc.

---

> ### Author Response · Authors · 2025-11-25
>
> We thank the reviewer for the thoughtful and encouraging assessment of our work and for recognizing the potential impact of our approach. Below we address the minor weakness and the reviewer’s question.
>
> **W1:** We appreciate the concern regarding ESWM’s applicability to continuous environments. To explore this, we conducted new experiments in a realistic 3D setting using ProcThor (Deitke et al., 2022), which provides continuous state and action spaces and substantial visual and structural variability. ESWM showed remarkable capacity to predict both actions and observations in this domain, despite the high-resolution inputs (224$\times$224$\times$3), the effectively unbounded action affordances, and the large diversity of environments (8,000 indoor scenes). After roughly 12 million samples, the model reached a cosine similarity of ~0.97 for state-embedding prediction, and 0.35 and 0.60 for displacement and rotation prediction respectively (chance: 0.015 and 0.027). We have added these results to Section 4.7 and Extended Methods C.1.4.
>
> **Q1:** We indeed plan to extend ESWM toward real-world embodiment. We plan to test ESWM on more realistic robot navigation settings that involve unstructured indoor spaces, where partial observability, sensor noise, and real-world dynamics provide ideal testbeds for evaluating ESWM’s generality. We believe the model’s compact latent state, its ability to support rollouts, and its efficiency in learning spatial structure make it a promising foundation for future embodied systems. We updated the Discussion section and the Impact Statement of our work in the Appendix to outline these steps.
>
>
> Deitke, M., VanderBilt, E., Herrasti, A., Weihs, L., Salvador, J., Ehsani, K., Han, W., Kolve, E., Farhadi, A., Kembhavi, A., & Mottaghi, R.  (2022). ProcTHOR: Large-Scale Embodied AI Using Procedural Generation. arXiv preprint arXiv:2206.06994.

---

### Official Review · Reviewer_xMiP · 2025-10-23

**Soundness:** 3
**Presentation:** 2
**Contribution:** 3
**Rating:** 6
**Confidence:** 5

**Summary:**

This paper introduces the Episodic Spatial World Model (ESWM), a framework that
constructs spatial representations from sparse, disjoint episodic memories. The
model is meta-trained to predict unseen transitions from minimal one-step
transition memories across diverse environments. The authors demonstrate that
ESWM forms geometric latent representations, enables zero-shot exploration and
navigation, and adapts rapidly to environmental changes.

**Strengths:**

1. Well-documented implementation: The paper provides comprehensive technical details including pseudocode (Algorithms 1-4), extensive appendices, and thorough experimental setup descriptions. This supports reproducibility.
2. Strong downstream performance: ESWM demonstrates impressive zero-shot capabilities on exploration (96.48% of oracle performance) and navigation tasks (96.8% success rate, 99.2% path optimality), outperforming the task-specific EPN baseline by substantial margins.
3. Interesting architectural insights: The finding that transformer attention mechanisms are critical for this task (while LSTM and Mamba struggle in Open Arena) provides valuable insights about the role of content-addressable memory in spatial reasoning.
4. Adaptability demonstration: The ability to handle environmental changes by simply updating memory banks (93% success rate with new obstacles) is a genuine advantage over weight-based world models.
5. Emergence of spatial structure: The spontaneous emergence of environment-like manifolds in latent space (Figures 3, 9-13) is compelling, especially the model's ability to infer spatial relationships from obstacle observations and boundary shapes.

**Weaknesses:**

## Major Concerns
1. Insufficient Justification of Core Premise.
The abstract and introduction claim that animals/humans build spatial maps from "disjoint experiences governed by consistent spatial rules," but no neuroscience evidence is provided to support this specific claim. The cited disruption studies (lesions, amnesia) show that MTL is important for spatial cognition and episodic memory, but don't demonstrate that humans actually construct maps from genuinely disjoint, fragmented experiences.
In reality, human and animal spatial experience is largely continuous and spatiotemporal. We don't teleport randomly through environments. The authors should either:
  * Provide evidence that biological systems actually face and solve this "disjoint memory" problem, or
  * Reframe their contribution as addressing a computational/AI challenge rather than claiming biological inspiration for this specific aspect

2. Unclear Relationship to Prior Work on Transition Systems.
The memory bank structure (graph of one-step transitions) closely resembles successor representations and transition graphs that have been extensively studied in both:
  * Hippocampal/entorhinal modeling: Stachenfeld et al. (2017), and related work on spatial transition representations in grid cells
  * General cognitive maps: Work from Behrens' lab on relational structure

The paper mentions Stachenfeld et al. only briefly and doesn't adequately distinguish ESWM's contribution from this and other related substantial body of work. How does ESWM's approach differ conceptually from learning successor representations or other transition models? This needs clear articulation.

3. Limited Scope and Generalization Concerns.
Environment Scale: All tested environments are very small (19-37 locations). With such limited state spaces and the model seeing many different configurations during meta-training, it's unclear whether ESWM is:
  * Learning generalizable spatial reasoning principles, or
  * Memorizing patterns across the finite set of possible small-scale configurations

The one generalization test (size-19 -> size-37) shows substantial performance drops (source->action->end accuracy: 59%->90%->55%), suggesting the model may not scale well.
Egocentric Setting: While MiniGrid experiments (Figure 8) show promise, they're presented quite briefly. Given that real-world spatial cognition is fundamentally egocentric, this deserves more thorough investigation.
Why hexagonal grids? The choice of hexagonal structure is never justified. This seems arbitrary and limits comparability with standard benchmarks.

4. Missing Statistical Rigor.
Many key results lack error bars or significance tests:
  * Figure 2: Training curves show variance bands, but test results don't
  * Figure 4a: No confidence intervals on entropy vs. path length
  * Figure 5: Some comparisons report significance, others don't
  * Exploration and navigation comparisons need statistical testing across multiple seeds
For a model claiming to learn generalizable principles, demonstrating statistical reliability is essential.

5. Insufficient Ablation Studies.
The paper would benefit from:
  * Ablating the "minimality" constraint more thoroughly: Figure 7 shows non-minimal banks work, but how does performance scale with memory bank size? Is there an optimal density?
  * Analyzing what makes transformers work here: Beyond showing LSTM/Mamba fail, what properties of self-attention are critical? Number of heads? Depth? Context length?
  * Testing robustness to noise: Real-world transitions are noisy—how sensitive is ESWM to observation or transition noise?


## Minor Concerns
1. Conceptual and Framing Issues.
Section 2.2's key question: "Can spatial maps also emerge in general-purpose models not explicitly trained for navigation?"
This has essentially been answered affirmatively by prior work (TEM itself, Behrens lab work, emergence in language models). The authors should clarify what new dimension their work adds to this question.

Section 2.1: The first paragraph of Related Work feels unfocused, jumping between human cognition, RL, and vision without clear connections.

2. Technical Clarifications Needed.
Graph theory terminology (Section 3.1): Please verify that "disjoint" is used consistently with standard graph theory definitions. In particular, "disjoint" typically refers to vertex or edge-disjoint subgraphs, but here seems to mean "non-sequential transitions". If that's the case, please state this somewhere.

"I don't know" classification: The handling of unobservable regions (17% of queries in Random Wall) is clever, but how is the threshold for outputting "I don't know" determined? Is it learned or hand-tuned?



## Minor Comments

* Line 124-125: "TEM factorizes sensory and structural information". Clarify how ESWM's approach differs
* Figure 1c: The masking procedure is clear, but consider showing an example with actual states/actions
* Section 4.4: The "zig-zag exploration strategy" is interesting. Can you characterize this more formally?
* Appendix B.2: The guided imagination approach using latent space geometry is valuable and could be elevated to the main paper

**Questions:**

1. Can you provide evidence that biological systems construct spatial maps from genuinely disjoint (not just sparsely sampled) experiences?
2. How does your approach differ fundamentally from learning successor representations or other graph-based transition models? What unique capability does ESWM provide?
3. Have you tested on substantially larger environments (100+ states)? What is the scaling behavior?
4. Why is transformer attention specifically required? Have you tried other relational reasoning architectures (e.g., Graph Neural Networks, which seem natural for transition graphs)?
5. Can you clarify the relationship between your work and existing transition system models, particularly for grid cells and hippocampal representations?

---

> ### Author Response · Authors · 2025-11-25
>
> **MajC1, Q1:** We appreciate the reviewer’s comment and clear suggestions. We also want to emphasize that our work was genuinely motivated by empirical neuroscience and by our understanding of how episodic memories are organized and reused in the MTL. Below we outline several lines of experimental evidence that, in our view, strongly support the core premise that animals and humans can integrate information collected across disjoint episodes into coherent spatial or relational maps.
>
> Our argument rests on two well-established observations. First, both rats and humans integrate information from episodic experiences that are separated across days. Second, this integration does not depend strictly on continuous spatiotemporal experience: episodic memories can be combined based on non-temporal factors such as goal demands, stimulus relationships, and contextual structure.
>
> Several behavioral studies make this explicit. Roberts et al. (2007) showed that rats inferred a shortcut between two locations in a maze after exploring its subcomponents in separate sessions on different days. From the perspective of our framework, the animals observed disconnected edges of the underlying spatial graph—never experiencing them as a continuous trajectory—yet were still able to infer the globally shortest route during the test. Fernandez et al. (2023) similarly demonstrated that humans trained to navigate locally within three separate environments on different days could integrate these fragmented experiences to navigate across them globally. Park et al. (2020) found that human subjects who learned each dimension of a 2D relational space in separate episodes nonetheless formed a joint map of the full space, again illustrating cross-episode integration.
>
> There is also substantial neural evidence that hippocampal and amygdala circuits integrate memories according to shared structure rather than temporal contiguity. Yokose et al. (2017) showed that amygdala-dependent memories from distinct tasks form overlapping engrams when they share a critical stimulus. Schlichting et al. (2015) found that the anterior hippocampus integrates across paired associates (A→B, B→C) to support the inferred A→C relationship. McKenzie et al. (2014) reported that hippocampal population codes become similar for objects that share spatial context, even when those objects were experienced in separate episodes. Related reviews (Morton et al. 2017; Brunec et al. 2020) synthesize a broad literature showing that the hippocampus organizes memories into integrated representations that reflect spatial, temporal, and conceptual structure—often beyond what was directly and continuously experienced.
>
> Altogether, these findings provide strong evidence that biological systems do face and solve a form of the “disjoint memory” problem. They routinely combine episodic fragments acquired at different times, in different contexts, and along different task dimensions, to construct coherent internal maps that support flexible behavior.
>
> We included an extended discussion to Appendix B of the revised paper to provide a better picture of the literature supporting the view that animals and humans can integrate information collected across disjoint episodes into coherent spatial or relational maps.

---

> > ### Author Response · Authors · 2025-11-25
> > **References**
> >
> > McKenzie, S., Frank, A. J., Kinsky, N. R., Porter, B., Rivière, P. D., & Eichenbaum, H. (2014). Hippocampal Representation of Related and Opposing Memories Develop within Distinct, Hierarchically Organized Neural Schemas. Neuron, 83(1), 202–215.
> >
> > Brunec, I. K., Robin, J., Olsen, R. K., Moscovitch, M., & Barense, M. D. (2020). Integration and differentiation of hippocampal memory traces. Neuroscience & Biobehavioral Reviews, 118, 196–208. https://doi.org/10.1016/j.neubiorev.2020.07.024
> >
> > Carpenter, F., Manson, D., Jeffery, K., Burgess, N., & Barry, C. (2015). Grid Cells Form a Global Representation of Connected Environments. Current Biology, 25(9), 1176–1182. https://doi.org/10.1016/j.cub.2015.02.037
> >
> > Fernandez, C., Jiang, J., Wang, S.-F., Choi, H. L., & Wagner, A. D. (2023). Representational integration and differentiation in the human hippocampus following goal-directed navigation. eLife, 12, e80281. https://doi.org/10.7554/eLife.80281
> >
> > Park, S. A., Miller, D. S., Nili, H., Ranganath, C., & Boorman, E. D. (2020). Map Making: Constructing, Combining, and Inferring on Abstract Cognitive Maps. Neuron, 107(6), 1226-1238.e8. https://doi.org/10.1016/j.neuron.2020.06.030
> >
> > Roberts, W. A., Cruz, C., & Tremblay, J. (2007). Rats take correct novel routes and shortcuts in an enclosed maze. Journal of Experimental Psychology: Animal Behavior Processes, 33(2), 79–91. https://doi.org/10.1037/0097-7403.33.2.79
> >
> > Schlichting, M. L., Mumford, J. A., & Preston, A. R. (2015). Learning-related representational changes reveal dissociable integration and separation signatures in the hippocampus and prefrontal cortex. Nature Communications, 6(1), 8151. https://doi.org/10.1038/ncomms9151
> >
> > Yokose, J., Okubo-Suzuki, R., Nomoto, M., Ohkawa, N., Nishizono, H., Suzuki, A., Matsuo, M., Tsujimura, S., Takahashi, Y., Nagase, M., Watabe, A. M., Sasahara, M., Kato, F., & Inokuchi, K. (2017). Overlapping memory trace indispensable for linking, but not recalling, individual memories. Science, 355(6323), 398–403. https://doi.org/10.1126/science.aal2690
> >
> > Morton, N. W., Sherrill, K. R., & Preston, A. R. (2017). Memory integration constructs maps of space, time, and concepts. Current Opinion in Behavioral Sciences, 17, 161–168. https://doi.org/10.1016/j.cobeha.2017.08.007

---

> ### Author Response · Authors · 2025-11-25
>
> **MajC2, Q2:** Successor representations and ESWM are similar in that both rely on transition information, but they differ in several important ways. First, classical SR methods require either a full transition matrix or incremental experience sufficient to estimate one. They are not designed to infer missing transitions or to complete a partially observed graph. ESWM, by contrast, is built specifically for that setting: it takes a sparse set of one-step episodic transitions and infers the missing structural relations directly at inference time. Conceptually, ESWM is estimating the entire family of transition distributions from incomplete data, whereas SR presupposes access to the full underlying dynamics.
>
> Second, SRs are learned through iterative, policy-dependent updates and encode multi-step, discounted future occupancy. ESWM does not rely on iterative updates and is not tied to a particular policy. The transitions in its memory bank are single, policy-free episodic observations, and the model performs transitive reasoning over these episodes on demand to answer arbitrary transition queries.
>
> In addition, SRs collapse the environment into a future-occupancy representation, which mixes structural and predictive information. ESWM keeps the structure explicit and manipulable, allowing it to reconstruct the transition graph itself rather than a discounted predictive summary (e.g. state values). ESWM is also symmetric in its inference capabilities: it jointly estimates forward transitions, backward transitions, and actions from the same memory bank, whereas SRs typically encode only forward predictive relationships.
>
> Finally, ESWM supports rapid structural updates by directly editing or adding episodic transitions, without needing to relearn a transition matrix. This contrasts with SRs and related models, which must re-estimate transition dynamics when the environment changes.
>
>
> Regarding work from the Behrens group (e.g., Whittington et al., 2020/2021, Bakersman et al. 2025), these models operate within the same family as TEM and assume access to sequentially sampled transition structure, backed by a learned path-integration scaffold. ESWM differs in that it makes no such structural assumptions and derives both sensory and structural predictions purely from sparse episodic transitions, as described in Section 2.1 and in our response to comment **MinComm1**.
>
> We added a new section in the Appendix (Appendix A) that provides a deeper discussion of the similarities and differences between ESWM and prior models including TEM and SR.

---

> ### Author Response · Authors · 2025-11-25
>
> **MajC3:** We understand the reviewer's concern about the possibility of memorization. We’d like to emphasize that our experiments are specifically designed such that memorization would not be feasible while our analyses also rigorously test on held out settings to assure true generalization. We emphasize that our results already strongly support that ESWM is learning generalizable spatial reasoning principles. First, although the environments are small, the factorization of location, states, walls, and memory bank structures makes the total number of possible configurations intractable (>10^33 possible environments alone; L192-193). Second, in Open Arena, the training and eval states are **completely separate**. The high validation accuracy (Fig.2) cannot be due to memorized patterns from training. Third, we have shown that ESWM builds a global spatial map in Fig.3,10,11,12,13,14 that is consistent under different memory bank structures observing the same environment (L1084-1086). Such results are unlikely if ESWM had simply memorized prediction-relevant, small-scale, local cyclic patterns. Lastly, we would like to point the reviewer to revised Section 4.7 and Extended Methods C.1.4 where we scaled ESWM to realistic 3D indoor scenes with high-dimensional observation, continuous egocentric actions, and large state space–further demonstrating the scalability of ESWM to much more challenging environments.
>
> **Regarding the generalization test to a larger environment (size 19 -> size 37) in Open Arena:** We agree that the performance drop seems large. However, considering the model is trained only in a fixed environment size (as opposed to Random Wall, where the model sees a distribution of environment size; L1018-1021), we consider this result to still be significant.
>
> **Regarding the MiniGrid experiment:** We appreciate the reviewer’s suggestion and agree that a deeper investigation of the egocentric agent trained in MiniGrid is highly relevant for understanding how ESWM scales to more realistic settings. Our new experiments in fully egocentric 3D environments already extend the scope of the paper and demonstrate that ESWM remains effective under more complex perceptual and dynamical conditions (see updated Section 4.7 and Extended Methods C.1.4).
>
> A more comprehensive analysis of the MiniGrid agent–including spatial selectivity, population-level geometry, and behavior under systematically varied layouts–is indeed valuable. However, the breadth of those analyses would significantly expand the manuscript and we believe they are better suited for a dedicated study. We therefore plan to pursue this line of work in a follow-up project, where we can examine these questions with the depth and clarity they deserve.
>
> **Regarding our choice of hexagonal layout:** on Lines 239-240  we had already mentioned our motivation for this choice which was to improve the state connectivity beyond traditional square grids.
>
> **MajC4:** We would like to first clarify that Fig.2 shows only test results, and we don’t have variance bands; the variations are due to iteration-to-iteration changes in performance.
>
> To address this comment we made several changes to the figures. 1) We replaced Fig4a’s STD with 95% confidence interval computed over all predictions for each path length. 2) We repeated the experiments in Fig.5 with 4 seeds of ESWM and EPN. We selected the best seed from ESWM and EPN for fig.5 and compared the mean performance of ESWM and EPN over different seeds in Fig.15.  A Welch’s two-sample t-test was used to compare ESWM and EPN ($\alpha$ = 0.05). We showed the significant differences between models with a dashed line with “*” ($p$ < 0.05).

---

> ### Author Response · Authors · 2025-11-25
>
> **MajC5:**
>
> **Regarding the minimality constraint:** This point was also brought up by Reviewer **3Ryj**. For brevity, we invite the reviewer to view our response to Reviewer 3Ryj’s **Q2** where we included a breakdown of performance over memory bank sizes from Fig.9 (originally Fig.7) and discussed the minimality constraints in more detail.
>
> We agree that it would be important to understand the factors leading to the Transformer’s success and failure of other architectures in Open Arena setting. In our response to Reviewer 3Ryj comment#**Q1**, we showed that LSTM and Mamba likely fail because they overfit on the training distribution of states and fail to generalize to new state values.
>
> **Regarding Transformer’s hyperparameters most affecting its performance:** We had already shown in Fig.2 deeper Transformers strongly boost performance. To further investigate this, we conducted additional experiments where we systematically changed the number of heads in our 6 layer model. We observe that while increasing depth leads to a monotonic increase across all three prediction tasks, increasing the number of heads, initially improves performance but exceeding 8 heads leads to quick drop in performance (see Table.1 in the revised manuscript or the table below). Importantly, Changing the number of heads drastically affects the state prediction tasks but only slightly affect the action prediction task. While non-conclusive, these results may suggest that while model depth impacts the general spatial reasoning capacity, the number of heads impacts accurate memory retrieval from past experiences that may be more crucial for state prediction.
>
> Regarding context length, we note that the memory banks have a fixed length across trials in Open Arena.
> ### Open Arena, ESWM-T-6L
>
> | Task            | 2 Heads | 4 Heads | **8 Heads** | 16 Heads |
> |-----------------|---------|---------|-------------|----------|
> | Predict Source  | 0.4049  | 0.7897  | **0.833**   | 0.4209   |
> | Predict Action  | 0.8461  | 0.9182  | **0.926**   | 0.9181   |
> | Predict End     | 0.3760  | 0.7811  | **0.796**   | 0.4047   |
>
>
>
>
> **Regarding robustness to noise**: To address this comment, we tested ESWM-T-4L under two types of noise in Random Wall setting:
>
>
> **a) Noise in state observations**. Where we incorporated memories with conflicting state observations at a particular location. To do this, we pick a location and change the corresponding memories in the memory bank to observe different states at that location.
>
>
> **b) Noise in transitions**. Where the same transition leads to contradictory outcomes. To do this, we randomly pick a transition from the memory bank, edit its end state, and append the modified transition to the memory bank.
>
> We observe that ESWM is more sensitive to noisy state observations compared with noisy transitions. The accuracies drop around 10% in the first case and around 1% in the second case, achieving ($s_s$ = 0.74, $a$ = 0.76, $s_e$ = 0.77) and ($s_s$ = 0.86, $a$ = 0.88, $s_e$ = 0.89) respectively (N=5000). These results suggest that ESWM has developed some robustness to noise, even though it is only trained on clean data. We added these new results in section 4.3
>
> **MinC1:** We agree that considering the prior work mentioned by the reviewer, our statement may be somewhat unspecific and its distinctness from those in prior work is not entirely clear. To address the reviewer’s concern, we revised this statement to the following: “Can spatial maps also emerge in general-purpose models not explicitly trained for navigation and operating entirely on a finite set of episodic memories?”
>
> **MinC2:** We revised section 2.1 to improve its flow and clarity.
>
> **MinC3:** Thank you for highlighting this potentially confusing point. It is indeed the case that we use “disjoint” to refer to non-sequential transitions. To avoid ambiguity and make this distinction clear, we state at line 174 that the memory bank contains disjoint transitions, meaning transitions that do not form a continuous trajectory. We also acknowledged that “disjoint” has also been used to refer to edge-disjoint subgraphs in section 4.2 and section D. We have revised those to use “disconnected” and reserve “disjoint” to exclusively mean non-sequential transitions.
>
> **Regarding I-don’t-know:** The “ I don’t know” classification is learned. It is treated as an additional output category during training, where the model needs to predict this specific class when receiving unsolvable queries.

---

> ### Author Response · Authors · 2025-11-25
>
> **MinComm1**: TEM separates the learning of structural and sensory information: its recurrent state captures a generic structural prior about how environments are organized, and this prior is embedded directly in the model’s weights. This design makes TEM very good at handling changes in sensory observations at familiar locations, but it also makes the model hard to adapt when the underlying structure changes or when entering environments with fundamentally different rules (e.g., RandomWall environments where walls can appear in arbitrary locations and shapes).
>
> In contrast, ESWM does not factorize these components. Both structural layout and expected sensory observations are inferred jointly and directly from episodic memories. Because ESWM does not rely on a fixed structural template encoded in its weights, it can rapidly integrate edits or additions to its memory bank and immediately update its predictions. This enables flexible, on-the-fly adaptation to new or altered environments without retraining.
>
>
> To clarify the differences between TEM and other models to ESWM, we added a new section to the appendix (Appendix A) where we now review these differences in detail.
>
> **MinComm2:** We thank the reviewer for the helpful suggestion. We added an example in Section 3.1, second paragraph.
>
> **MinComm3:** Our grid environment is a triangular lattice. We observe that the exploration paths visit at most two out of the three edges in any triangle, inferring the spatial relations of all three states using the minimal transitions. We also observe that the agent only visits three out of the four edges in a parallelogram (two triangles stitched together), forming a bigger zig-zag pattern.
>
> **MinComm4:** We agree with the reviewer that these results are indeed interesting and should ideally be included in the main paper. However, we were unable to fit these results in the main paper due to the volume of the work that we intended to present within a 10-page limit. We are open to any suggestions from the reviewer to better structure the paper.
>
> **Q3:** We investigate ESWM’s scalability through the MiniGrid experiment and the newly included ProcThor experiment (see the updated section 4.7 and Extended Methods C.1.4). In MiniGrid, the agent sees a 5$\times$5 view, observing 5$\times$5 = 25 locations at any particular position $\times$ heading. Each location can be associated with one of the 9 possible items. The theoretical maximum of the number of unique states is >7*10^23. However, the agent's limited view causes different position $
> times$ heading pairs to produce identical observations. Empirically, we observe a total of 9m unique states across 100,000 environments.
>
> In ProcThor, both positions and rotation are continuous – positions are continuous 2d coordinates while rotation is a continuous value between 0 and 360. Coupled with objects and room configuration, the number of states is intractable.
>
>
> **Q4:**
> We sincerely appreciate the reviewer’s insightful suggestion regarding GNNs as another possibly relevant method to our work. While we agree that this is an interesting direction, thoroughly implementing and analyzing this approach would require substantially more time than is available during the rebuttal period. We therefore hope the reviewer understands that we are unable to include this experiment at this stage.
>
> **Q5:** We believe the reviewer is inquiring about the relationship with methods such as successor representation to which we have provided a response under **MajC2**. However, we remain open to further discussions, if the reviewer had meant otherwise.

---

> > ### Comment · Reviewer_xMiP · 2025-11-25
> >
> > I thank the authors for their thorough rebuttal and additional information they add to their manuscript. I also thank them for pointing me to the details that I missed in my review, especially those which were already part of their submission and stand corrected. In light of the amount of improvements, I am happy to increase my score.

---

### Official Review · Reviewer_3Ryj · 2025-10-28

**Soundness:** 3
**Presentation:** 3
**Contribution:** 2
**Rating:** 4
**Confidence:** 4

**Summary:**

This paper proposes the Episodic Spatial World Model (ESWM), a framework for building coherent spatial world models from **sparse and disjoint one-step transitional memories**. Inspired by the dual role of the Medial Temporal Lobe (MTL) in processing both episodic memory and spatial information, ESWM infers the underlying structure of an environment from a 'memory bank'—a set of independent transitions ($s_s​,a,s_e$​)—unlike prior models that rely on continuous trajectories.

The authors demonstrate the efficacy of ESWM through experiments in various grid-world environments (Open Arena, Random Wall, and MiniGrid). The results show that ESWM achieves higher prediction accuracy than the sequence-based model, TEM-T, and exhibits strong generalization to environments with unseen structures. Through ISOMAP visualizations, the authors reveal that ESWM's latent space forms a geometric map that accurately reflects the physical topology of the environment, including obstacles. This learned model enables high performance on downstream tasks such as zero-shot exploration and navigation. Furthermore, by decoupling memory from reasoning, the proposed architecture allows for rapid adaptation to environmental changes by simply updating the memory bank without retraining the model.

The paper's claims are well-supported by a detailed appendix that enhances reproducibility. It includes specifics on the model architecture, data generation pipeline, and additional results, such as the extension to the high-dimensional, vision-based MiniGrid environment (Figure 8), providing substantial evidence for the method's effectiveness.

**Strengths:**

- **Important Research Question and Interdisciplinary Connection:** This paper addresses a significant limitation of prior world models: their difficulty in inferring robust spatial knowledge from continuous sequences and adapting to environmental changes. The authors propose the Episodic Spatial World Model (ESWM), a novel approach inspired by the dual role of the Medial Temporal Lobe (MTL) in processing both episodic and spatial memory. This represents an excellent example of leveraging insights from neuroscience to advance machine learning research.

- **Comprehensive and In-depth Experiments:** The authors validate their claims through a diverse and rigorous set of experiments. By evaluating prediction accuracy in various environments (Open Arena, Random Wall), visualizing the latent space with ISOMAP, and demonstrating strong performance in zero-shot exploration and navigation, the paper not only shows that ESWM is effective for spatial reasoning tasks but also provides intuitive evidence for why it works.

- **High Reproducibility and Detailed Analysis:** The manuscript enhances the credibility and reproducibility of its results by providing meticulous details on model architectures and experimental setups in both the main paper and the appendix. The authors offer specific descriptions of the query generation (Appendix A.4), the ISOMAP visualization process (Appendix A.6), and provide extensive additional analyses (Appendix B) and visualizations (Appendix C), which lend strong support to their findings.

**Weaknesses:**

- **Insufficient Discussion of Closely Related Work and Unclear Novelty:** The core mechanism of the proposed method shares significant similarities with Generative Temporal Models with Spatial Memory (GTM-SM) [1], particularly the idea of leveraging one-step transitions to build a spatially-aware model. GTM-SM also demonstrated that its inferred state representations align with the true geometry and enable high-quality long-term predictions (e.g., Figure 2 in [1]). However, this paper lacks any discussion comparing ESWM with GTM-SM. I believe a thorough comparison is crucial for contextualizing the contributions of this work and clarifying its novelty. I strongly encourage the authors to incorporate this discussion.

- **Limited Evaluation Outside of Grid-World Environments:** While the authors test their model in several environments, all of them are grid-based. The discrete actions and cyclic nature of grid-worlds may provide implicit structural cues that simplify the task of learning spatial relationships. It is unclear whether ESWM's strong performance would generalize to more realistic, continuous environments (e.g., DMLab [2]) where such cues are absent. The lack of experiments in these settings makes it difficult to assess the true robustness and scalability of the proposed method.

- **Compliance with LLM Usage Policy:** The authors do not explicitly state whether or how they used Large Language Models (LLMs) in preparing their submission, as required by the ICLR 2026 Author Guide.

- **Minor Presentation Issues:**
    - In Figure 1a (middle), the caption "Specific parts of the environment may be changed dynamically" could be misinterpreted as changes occurring mid-trajectory. It would be clearer to state that wall configurations are randomized across different episodes.
    - In Figure 2, the font size is too small, and the x-axis labels and ticks are occluded. In the caption, `Open Arena` should be $\texttt{Open Arena}$, and the formatting for ”-T” is inconsistent.
    - In Figure 3, the font sizes are too small for easy reading.

[1] Fraccaro, Marco, et al. "Generative temporal models with spatial memory for partially observed environments." _International conference on machine learning_. PMLR, 2018.

[2] Beattie, C., Leibo, J. Z., Teplyashin, D., Ward, T., Wainwright, M., Kuttler, H., Lefrancq, A., Green, S., Vald ¨ es, ´ V., Sadik, A., Schrittwieser, J., Anderson, K., York, S., Cant, M., Cain, A., Bolton, A., Gaffney, S., King, H., Hassabis, D., Legg, S., and Petersen, S. DeepMind Lab. CoRR, abs/1612.03801, 2016.

**Questions:**

- In the Figure 2 results, the performance of Mamba and LSTM architectures is notably poor in Open Arena but significantly better in Random Wall. This is counter-intuitive, as the Random Wall task seems more complex due to obstacle wall. Could the authors provide an explanation for this performance discrepancy?

- Regarding the experiments on non-minimal memory banks (lines 293-295), you show that ESWM-T performs well. Does this finding hold for other architectures like ESWM-Mamba and ESWM-LSTM, as well as for the TEM-T baseline? Additionally, could you elaborate on why a duplicated memory item might be problematic? Is it because it reduces the diversity of transitions sampled within a finite memory capacity?

- In Appendix A.4, you describe three different query types used for the Random Wall experiments (unseen, seen, and unsolvable). Could you please provide a performance breakdown for each query type? I suspect this data might help explain why ESWM-Mamba and LSTM perform better in the Random Wall setting.

---

> ### Author Response · Authors · 2025-11-25
> **Response to weaknesses**
>
> **W1: Relation to GTM-SM.** Thank you for pointing us to this paper. We were unaware of this paper and agree that it is highly relevant to our work and predates some of the other models such as TEM. GTM-SM is structurally and functionally similar to TEM-t (Whittington et al. 2022) where a recurrent network (a state-space model in GTM-SM) updates its internal state according to the sequence of actions taken by an agent, and this internal state is used as key/query in an external memory where each key is paired with the embedding of the observations. Similar to TEM-t, as the structural information of the environment is embedded in a state space model’s weights, GTM-SM faces similar limitations as TEM. In particular, it cannot adapt to structural changes to the environment without extensive training of the state-space model weights (also see our response to Reviewer **xMiP**’s comment #**MinComm1** on the differences between TEM and ESWM). We updated the last paragraph of section 2.1 and discussed the main differences between GTM-SM and ESWM in detail.
>
> **W2: Limited evaluation.** To address the reviewer’s concern, we scaled ESWM to a procedurally-generated, realistic 3D indoor scene dataset – ProcThor (Deitke et al. 2022). This environment consists of high-resolution images containing a wide variety of objects and continuous state and action spaces. We trained ESWM on 8000 procedurally generated indoor scenes and found that it succeeds at solving this challenging task by learning to predict both the state vectors and action parameters from high resolution pixel inputs. Importantly, the memory bank remains compact, validating the scalability of ESWM to much larger environments. Please see the updated Section 4.7 and  Extended Methods C.1.4 for more details.
>
> **W3:  LLM Usage.** To our knowledge, the ICLR 2026 Author Guide only mandates the usage to be declared only “*if* LLMs play a significant role …”. We opted not to do so since our usage of LLMs was strictly limited to word editing and it was not involved in any other aspects of our work. To address the reviewer’s concern, we added a statement about our specific use of LLMs in preparing the manuscript to be fully transparent about this.
>
> **W4: Minor Presentation Issues.** We thank the reviewer for the valuable feedback. We have modified Fig.1 and Fig.2 according to the reviewer’s suggestions in the revised manuscript. We will revise the font size in Fig.3 and provide an updated manuscript by Dec 3.

---

> ### Author Response · Authors · 2025-11-25
> **Response to questions**
>
> **Q1:** We understand the confusion and are happy to clarify this point for the reviewer. First, a key difference between the two tasks is that *Open Arena uses mutually exclusive training and validation states,* whereas Random Wall does not (L244–245). This means that during evaluation in Open Arena, the model is tested on completely unseen states. Looking closely at the models’ performances shown in the new Fig.8, we observe that LSTM and Mamba overfit to the training states while the Transformer model generalizes in Open Arena.
>
> Second, we note an additional observation: *LSTM struggles to perform even on the training set in Open Arena*, unlike in Random Wall where it succeeds on both train and validation splits. We suspect another task difference contributes to this, States in Open Arena consist of a 6-bit value, each represented by a 128-dimensional vector(see L915-917) while states in Random Wall consist of a single vector. The latter may allow a more compact latent representation that better fits within LSTM’s limited memory capacity. We added a discussion of these observations to Appendix D.1.
>
> **Q2:** We additionally tested the Mamba and LSTM variations of the model and updated Fig.9 to include ESWM-Mamba-5L , ESWM-LSTM-6L, and TEM-T-2L on non-minimal memory banks in Open Arena. We find their results to be on a par with training with minimal memory.
>
> We agree that the drop in performance for ESWM-T-12L trained on non-minimal memory banks is counterintuitive. To investigate why the model underperforms while having access to more experience in an environment, we break down the model’s performance across memory bank sizes and observe a substantial gap between denser and sparser memory banks, even though they are equally likely to be in the training data (Fig.9b)). This suggests that, when trained on a mixture of densities, the model focuses more on solving the easier tasks where it is given an overcomplete memory bank, and only partially succeeds in learning the more challenging tasks when minimal information is given. This leads to weaker spatial reasoning skills and poorer performance when the observation pattern changes.
>
> In contrast, ESWM-T-12L trained under minimality constraints is forced to solve the harder spatial reasoning problem consistently. As shown in the revised Fig.4c), this produces representations that transfer robustly to denser, Out-of-Distribution (OOD) memory banks. We also note that within the training distribution (ID), increasing memory bank size corresponds to an expansion of the observed area, which requires the model to coherently build a larger map which is more challenging and likely explains the performance drop.
>
> Collectively, these findings suggest that applying sparsity constraints during training leads to better test-time flexibility in terms of information availability.
>
> We added a discussion of these results to Section D.2.
>
> **Q3:** We thank the reviewer for this insightful comment. We reanalyzed the models predictions and generated  the performance breakdown for ESWM-T-4L, ESWM-LSTM-4L, ESWM-MAMBA-5L on the three query types (N=1000) (shown in the Table below and in the revised manuscript Table.2, Table.3, Table.4). These results show that, while seen and unsolvable query types’ performance is roughly comparable across model types, the performance for unseen queries varies significantly between models. This indicates that the performance gap in unseen queries is the main contributor to the performance gap between models.
>
> ### ESWM-T-4L
>
> |              | Predict Source | Predict Action | Predict End |
> |--------------|----------------|----------------|-------------|
> | **Seen**     | 0.9050         | 0.8700         | 1.0000      |
> | **Unseen**   | 0.8750         | 0.8880         | 0.9030      |
> | **Unsolvable** | 0.8270       | 0.9050         | 0.8180      |
>
> ### ESWM-LSTM-4L
>
> |              | Predict Source | Predict Action | Predict End |
> |--------------|----------------|----------------|-------------|
> | **Seen**     | 0.8780         | 0.8690         | 1.0000      |
> | **Unseen**   | 0.6660         | 0.7530         | 0.7030      |
> | **Unsolvable** | 0.7620       | 0.8830         | 0.7560      |
>
> ### ESWM-Mamba-5L
>
> |              | Predict Source | Predict Action | Predict End |
> |--------------|----------------|----------------|-------------|
> | **Seen**     | 0.9000         | 0.8810         | 1.0000      |
> | **Unseen**   | 0.7710         | 0.8110         | 0.8230      |
> | **Unsolvable** | 0.8110       | 0.8990         | 0.8010      |

---

> > ### Comment · Reviewer_3Ryj · 2025-11-25
> >
> > Thank you for the detailed responses to my concerns and for conducting additional experiments within the limited rebuttal period. In particular, I appreciate the clear and concrete explanation of how your method relates to the prior work Generative Temporal Models with Spatial Memory (GTM-SM), as well as the new results in non-grid environments addressing my scalability concerns.
> >
> > Based on these clarifications and additional evidence, I am raising my score and leaning toward acceptance.

---

### Official Review · Reviewer_d6hX · 2025-10-28

**Soundness:** 3
**Presentation:** 4
**Contribution:** 4
**Rating:** 8
**Confidence:** 4

**Summary:**

The paper proposes a novel framework Episodic Spatial World Model (ESWM) for constructing cognitive maps from disjoint episodic memories, akin to human learning behavior. Instead of using long consecutive trajectories over the environment, ESWM uses a meta-learning algorithm across many environments to infer unseen transitions from a sparse memory bank of one-step experiences, learning spatial regularities on the way. Experiments demonstrate useful properties of the framework, including the ability to learn an internal representation that closely mirrors the environment layouts with minimal observations, and being sample-efficient and adaptive.

**Strengths:**

1. The empirical evaluation is thorough, rigorous, and insightful. The authors test ESWM on multiple tasks and environment variants (open arenas vs. maps with random walls/obstacles) and over multiple base models (transformers, LSTM, and Mamba) to assess its prediction accuracy, representation quality, and control performance. The set of experiments covers many insightful explorations of the ESWM behavior, including memory integration, navigation, adaptation to environment change, and scaling. Open Arena training also reveals nontrivial performance difference between the transformer ESWM and the other two variants.
2. This work has substantial implications for both reinforcement learning and cognitive modeling. By showing that an agent can rapidly form a useful world model from a handful of potentially disjoint experiences, it highlights a pathway toward more generalizable and lifelong learning in AI. In practical terms, the ability to plan efficient exploration and navigation with less training could be impactful for robotics or embodied AI, where an agent might get only sparse observations of a new environment before needing to act.

**Weaknesses:**

1. The innovation is a bit limited, and there is no baseline outside of the ESWM model to compare with, making it hard to ground the model properties in context. Given the nature of the base models and the environment, the key innovation seems to be the training scheme over multiple memory banks. On the grounding, it would be interesting to see how the model performs against ordinary baselines or training schemes. For example, one may compare ESWM-T with ordinary transformers (with a single memory bank). Comparing ESWM-T with a model trained with longer trajectories would also provide insight into how sparse episodic memory effects learning.
2. While the results are strong, the experiments are confined to relatively simple, synthetic environments. Additionally, the environments considered all follow consistent spatial rules. If an environment had more complex dynamics or non-grid connectivity, it’s unclear if the current approach would generalize as well (beyond scale).

**Questions:**

1. Since ESWM essentially learns to infer spatial structure, have you considered comparing it against a non-learned mapping baseline?
2. How crucial is it for the memory bank to form a perfect spanning tree of the environment? In cases where the episodic memories are incomplete, how might ESWM perform or adapt?

---

> ### Author Response · Authors · 2025-11-25
> **Response to weaknesses**
>
> We’re encouraged to see the reviewer finds our work thorough, insightful, and impactful, and for having recognized its implications in reinforcement learning, and cognitive modeling. Below, we provide a point-by-point response to the reviewer's comments.
>
> **W1:** We appreciate the reviewer’s suggestion and would like to clarify two points. First, meta-training across environment $\times$ memory bank pairs has been used in prior work (e.g., TEM), and we do not claim this as our main contribution. Instead, the novelty of our work lies in the instantiation of the memory bank as a set of temporally and spatially independent transitions. This design offers scalability by sparsely incorporating experiences from different episodes, and adaptability through memory editing. Second, ESWM is trained over numerous memory banks sampled from **different** environments (L187) and as a result it can generalize to any environment unseen during training. A model with a fixed memory bank therefore, cannot be expected to generalize to unseen environments.
>
> Regarding the suggestion about comparison with “model trained with longer trajectories”, we interpreted the reviewer’s suggestion as investigating how ESWM behaves when trained with non-minimal memory banks. We ran additional experiments to study this in two settings:
>
> **1) Adding uniformly sampled disjoint memories** to the minimal memory banks, resulting in non-minimal, yet still disjoint, memory banks ranging between minimal spanning tree (MST) and a complete graph. As shown in Fig.9, ESWM learns effectively under these variations.
>
>  **2) Training on continuous trajectories larger than MST**, in which we increase the number of memories by +5 (Open Arena) and +14 (Random Wall) on average. This setup matches the memory bank used for the TEM-T baseline. Under this setting, ESWM-T-10L achieves final accuracies of ($s_s$ = 0.99, $a$ =0.99, $s_e$ =0.99) in Open Arena, slightly higher than the one trained with minimal memory banks. In Random Wall, ESWM-T-6L’s start state and action prediction accuracy remain mostly unchanged, while the end state prediction’s accuracy drops by 15% ($s_s$ = 0.81 , $a$ = 0.89, $s_e$ =0.78). Collectively, these results demonstrate the ESWM’s robustness across memory bank structures and density. We added a discussion of these results in Appendix D.1 and remain open to any other suggestions.
>
> **W2:** We agree the MiniGrid and hexagonal grid environments are both relatively simple and it’s unclear whether the model can be scaled to more complex scenarios. Yet, we believe some of our current results already provide positive evidence regarding environments with *non-grid connectivity* and *complex dynamics*. We interpreted “grid connectivity” as action affordances that uniformly cover all directions within every location and “complex dynamics” as state-dependent action affordances. If that was the intended meaning by the reviewer, then environments such as Random Wall and MiniGrid already possess non-grid structures in that sense, and contain state-dependent action affordances that vary across environments. ESWM is evaluated on such a distribution of the environment in Fig.2) Random Wall, and ISOMAP analysis also shows a coherent spatial map under such partial observation (Fig.3c, d; Fig.12b).
>
> To address this request further, we ran additional experiments in complex, realistic 3D settings generated using the ProcThor environment which further demonstrated the scalability of the proposed approach to those settings (see the implementation details in Section 4.7 and Extended Methods C.1.4). ESWM succeeded in predicting both states and actions even when inputs consisted of high-resolution images rendered within a 3D virtual environment with highly flexible egocentric action affordances. We believe that these results together provide strong support in favor of ESWM’s scalability. However, we remain open to any further suggestions from the reviewer, if these experiments do not fully cover their intent.

---

> ### Author Response · Authors · 2025-11-25
> **Response to questions**
>
> **Q1:** We were unsure about what the reviewer precisely meant by “non-learned mapping baseline” and interpreted that as quantifying the resemblance between ESWM’s internal spatial map and its external environment using an established metric. To address this, we computed the Spatial Representational Similarity Analysis (sRSA) (Levenstein et al., 2024), which measures the correlation between latent and physical distances, on ESWM-T-2L across N = 1000 environments (5 memory banks from each environment). The mean sRSA is 0.78+-0.02, which is significantly higher than that of an untrained ESWM-T-2L, which has a mean of 0.50 +-0.04 (paired t-test, t=172.17, p < 1*10^-3). These results further validate that ESWM indeed organizes its internal representation to match its external spatial layout. We added these results to Section 4.2 and Extended Methods C.8. We remain open to further discussions if the reviewer meant otherwise.
>
> **Q2:** We investigated the reviewer’s question along two axes:
>
> **When the memory bank contains more memories than the minimal spanning tree (MST)**. For these results, please refer to our response to **W1**.
>
> **The memory bank contains fewer memories than the MST**. We want to first clarify that, as described in L208-212, L1019-1020, ESWM training in Random Wall is not solely done on the full environment MST, but also includes trials containing random subgraphs, where only a subset of all nodes is observed. In fact, the accuracies reported in Fig.2b display the average across all such conditions. For a detailed analysis of performance given various memory bank sizes, see our response to Reviewer **3Ryj** comment **#Q2** and Fig.4c in the revised paper. Moreover, we evaluated ESWM’s prediction performance on the memory banks collected by the exploring agent from section 4.4, which visits nodes along continuous trajectories, and observed a robust performance ($s_s$ = 80.9 , $a$ = 86.5, $s_e$ = 89.4). Importantly, ESWM’s ability to reason with incomplete memory banks to determine unknown regions is the core mechanism that underpins the success of our exploration agent.
>
> Daniel Levenstein, Aleksei Efremov, Roy Henha Eyono, Adrien Peyrache, Blake Richards bioRxiv 2024.04.28.591528; doi: https://doi.org/10.1101/2024.04.28.591528

---

### Author Response · Authors · 2025-12-01
**Rebuttal summary**

To facilitate the AC’s meta review, we summarize below the primary concerns raised by reviewers and the steps we took to address them in the revision.

**Scalability to high-dimensional continuous environments.**
We conducted new experiments in realistic, high-dimensional 3D continuous environments using ProcThor, demonstrating ESWM's scalability to these settings to predict both states and actions. These results address reviewer **d6hX**’s weakness #2, **3Ryj**’s weakness #2, **xMiP**’s major concern #3, and **o4ni**’s minor concern #1 and question #1. The new experiments were added to Section 4.7 and Extended Methods C.1.4.

**Clarifying the relation to prior work and neuroscience evidence for the core premise.**
We substantially revised Section 2.1 to clarify ESWM’s relationship to GTM-SM and added a new Section A in the appendix discussing connections to Successor Representation, the Tolman–Eichenbaum Machine, and recurrent predictive models. We also added a new section B in the appendix to provide an in depth review of the behavioral and neural evidence supporting our core premise: "building spatial maps from disjoint experiences". These changes address reviewer **3Ryj**’s weakness #1 and **xMiP**’s major concerns #1-2 and questions #1-2.

**Additional ablation studies.**
We also ran several ablation experiments and additional analyses. We trained additional ESWM and TEM-T models on non-minimal memory banks and performed new analyses on these checkpoints. We also trained ESWM variations with different number of heads to isolate its effect. These results address **d6hX**’s weakness #1 and question #2, **3Ryj**’s question #2, and **xMiP**’s major concern #5 (point #1). The new findings are included in Fig. 4c, Fig. 9, Section D.1 (last paragraph), and Section D.2.

In addition, we carried out other reviewer-suggested experiments investigating the model robustness to memory noise and generalization. We incorporated these new findings and further clarifications into the revised manuscript to address the remaining points.

Overall, we carried out 11 new experiments, added or updated 8 figures, added 4 new tables, and added 3 new sections to the appendix (A, B, and C.1.4). We believe these revisions significantly strengthened the paper both conceptually and empirically.

---

### Meta-Review · Area_Chair_Ct2P · 2025-12-28

**Summary:**

The paper proposes ESWM, a neuroscience-inspired model that builds geometrically accurate and adaptive cognitive maps from sparse, disjoint episodic memories, enabling near-optimal exploration and navigation without additional training.

Across reviewers, the major concerns center on unclear novelty and positioning of ESWM relative to prior transition-based and spatial memory models, insufficient grounding in neuroscience and existing theory, and a lack of comparisons to standard baselines. Reviewers also consistently questioned generalization and scalability, as experiments were largely limited to small, grid-world environments, raising doubts about robustness to continuous, larger-scale, or noisier settings. Additional concerns included insufficient ablations and statistical rigor, as well as clarity and presentation issues.

After the rebuttal, the authors did a good job addressing almost all major concerns by adding extensive new experiments demonstrating ESWM’s scalability to high-dimensional continuous environments, clarifying its relationship to prior models and neuroscience evidence, and conducting additional ablations on memory structure and architecture.

Given the current status of the paper, AC recommends acceptance.

**Reviewer Concerns:**

Reviewer d6hX: The reviewer questions the novelty and grounding of ESWM, noting the lack of comparisons to standard baselines (e.g., ordinary transformers, longer-trajectory training, or non-learned mapping methods), and raises concerns about generalization beyond simple, grid-like synthetic environments with consistent spatial rules, as well as the model’s robustness to incomplete or imperfect episodic memory coverage.

Reviewer 3Ryj: The reviewer questions the novelty and positioning of ESWM, emphasizing strong conceptual overlap with prior work such as GTM-SM, which is not discussed or compared against. They also raise concerns about limited evaluation, as all experiments are confined to grid-world environments with discrete actions, leaving generalization to continuous or more realistic environments unclear. Additional issues include missing compliance statements on LLM usage, several presentation and clarity problems, and requests for deeper analysis of performance discrepancies (e.g., Mamba/LSTM behavior), memory bank duplication effects, and finer-grained result breakdowns in the Random Wall setting.

Reviewer xMiP: This reviewer raises substantial conceptual and methodological concerns, arguing that the paper’s biological motivation is insufficiently justified, particularly the claim that animals construct spatial maps from genuinely disjoint experiences. They also find the relationship to successor representations and transition-based cognitive map models underdeveloped, with ESWM not clearly distinguished from existing theories. Further concerns include limited environment scale, potential memorization rather than generalization, lack of statistical rigor, and insufficient ablation studies (e.g., memory density, noise robustness, architectural necessity of transformers).

Reviewer o4ni: The reviewer expresses a minor concern about generalization, noting that the proposed approach may not extend naturally to continuous or real-world environments.

After the rebuttal, the authors addressed most of the major concerns raised by the reviewers summarized above through additional experiments, clearer positioning relative to prior work, and more thorough analyses.

**Reviewer Scores:**

The initial scores were 8, 8, 6, and 4; after the rebuttal, Reviewer xMiP indicated an increase from 6 to 8 and Reviewer 3Ryj indicated an increase from 4 to 6, while the remaining two reviewers did not update their scores but had already been positive initially.

---

### Decision · Program_Chairs · 2026-01-26

Accept (Poster)